# The epigenetic regulator Mll1 is required for Wnt-driven intestinal tumorigenesis and cancer stemness

Johanna Grinat [1,11], Julian Heuberger [1,10,11 ✉], Ramon Oliveira Vidal [2], Neha Goveas [3], Frauke Kosel[1], Antoni Berenguer-Llergo [4], Andrea Kranz[3], Annika Wulf-Goldenberg[5], Diana Behrens[5], Bálint Melcher[6], Sascha Sauer [2], Michael Vieth[6], Eduard Batlle[7,8,9], A. Francis Stewart[3] & Walter Birchmeier [1 ✉]

Wnt/β-catenin signaling is crucial for intestinal carcinogenesis and the maintenance of intestinal cancer stem cells. Here we identify the histone methyltransferase Mll1 as a regulator of Wnt-driven intestinal cancer. Mll1 is highly expressed in Lgr5[+] stem cells and human colon carcinomas with increased nuclear β-catenin. High levels of *MLL1* are associated with poor survival of colon cancer patients. The genetic ablation of Mll1 in mice prevents Wnt/β-catenin-driven adenoma formation from Lgr5[+] intestinal stem cells. Ablation of Mll1 decreases the self-renewal of human colon cancer spheres and halts tumor growth of xenografts. Mll1 controls the expression of stem cell genes including the Wnt/β-catenin target gene *Lgr5*. Upon the loss of Mll1, histone methylation at the stem cell promoters switches from activating H3K4 tri-methylation to repressive H3K27 tri-methylation, indicating that Mll1 sustains stem cell gene expression by antagonizing gene silencing through polycomb repressive complex 2 (PRC2)-mediated H3K27 tri-methylation. Transcriptome profiling of Wnt-mutated intestinal tumor-initiating cells reveals that Mll1 regulates Gata4/6 transcription factors, known to sustain cancer stemness and to control goblet cell differentiation. Our results demonstrate that Mll1 is an essential epigenetic regulator of Wnt/β-catenin-induced intestinal tumorigenesis and cancer stemness.

[1] Cancer Research Program, Max Delbrück Center for Molecular Medicine (MDC) in the Helmholtz Society, 13125 Berlin, Germany. [2] Laboratory of Functional Genomics, Nutrigenomics and Systems Biology, Scientific Genomics Platforms, Max Delbrück Center for Molecular Medicine (BIMSB/BIH), 13092 Berlin, Germany. [3] Biotechnology Center, Center for Molecular and Cellular Bioengineering, Technische Universität Dresden, 01307 Dresden, Germany. [4] Biostatistics and Bioinformatics Unit, Institute for Research in Biomedicine (IRB Barcelona), The Barcelona Institute of Science and Technology, Barcelona, Spain. [5] Experimental Pharmacology & Oncology (EPO), 13125 Berlin, Germany. [6] Institute for Pathology, Klinikum Bayreuth, 95445 Bayreuth, Germany. [7] Institute for Research in Biomedicine (IRB Barcelona), The Barcelona Institute of Science and Technology, Barcelona, Spain. [8] ICREA, Institució Catalana de Recerca i Estudis Avançats, Barcelona, Spain. [9] Centro de Investigación Biomédica en Red de Cáncer (CIBERONC), Barcelona, Spain. [10] Present address: Division of Gastroenterology and Hepatology, Medical Department, Charité University Medicine, 13353 Berlin, Germany. [11] These authors contributed equally: Johanna Grinat, Julian Heuberger. ✉email: julian.heuberger@charite.de; wbirch@mdc-berlin.de

Wnt/β-catenin signaling controls the self-renewal of adult stem cells and promotes oncogenesis[1–4]. Activated Wnt/β-catenin signaling is associated with colon cancer and many other human cancers[5–7]. The inactivation of the destruction complex composed of APC, Gsk3β, and Axin1/2 stabilizes β-catenin, which would otherwise be phosphorylated and ubiquitinated for proteasomal degradation[8,9]. Stabilized β-catenin translocates to the nucleus, where it binds to Tcf/Lef1 transcription factors and induces the expression of target genes such as Axin2[10,11]. Mutations in Wnt signaling components initiate intestinal tumorigenesis by promoting the aberrant stability and hyperactivity of β-catenin[12,13].

Adult intestinal stem cells have been identified as the cells-of-origin of intestinal cancer[14]. Actively proliferating stem cells, which were initially described as crypt base columnar (CBC) cells, reside in stem cell niches located at the base of intestinal crypts. These cells constantly produce progenitor cells that differentiate into absorptive enterocytes and secretory cells, including Paneth, goblet, enteroendocrine, and tuft cells[15]. Inverse gradients of Wnt and Bmp signaling organize the intestinal epithelium in proliferative crypt and differentiated villus compartments. The intestinal stem cells express the leucine-rich repeat-containing G-protein coupled receptor Lgr5, which was discovered in the colon cancer cell line Ls174T[16,17]. High Wnt/β-catenin signaling in the crypts induces Lgr5 expression and is crucial for stem cell maintenance[17–19]. Aberrant Wnt signaling in the intestinal stem cells expands the stem cell compartment and initiates tumorigenesis[14].

Oncogenic Wnt signaling activates a stem cell gene expression program in colon cancer cells and confers cancer stemness[20]. Current therapies for colon cancer are often ineffective because cancer stem cells can resist conventional chemotherapy and initiate tumor relapse and metastasis. Epigenetic therapies hold great promise for inhibiting tumor growth and cancer stemness[21]. Oncogenic Wnt signaling has been shown to drive salivary gland and head and neck tumorigenesis by exploiting the epigenetic regulator Mll1[22,23]. Mll1 is a trithorax homolog histone methyltransferase of the Mixed lineage leukemia (Mll/Kmt2) family, which tri-methylates histone 3 at lysine 4 (H3K4me3), a chromatin mark that is enriched at the transcription start sites (TSS) of active genes. H3K4 tri-methylation antagonizes gene repression through H3K27 tri-methylation by polycomb complexes[24]. A role for Mll1 in Wnt-driven colon cancer has not yet been determined.

In this work, we identify the histone methyltransferase Mll1 as an epigenetic regulator of human and mouse intestinal cancer stem cells and tumors. Mll1 promotes β-catenin-induced intestinal stem cell expansion and tumorigenesis, and controls the self-renewal of colon cancer stem cells. Cancer stem cells are depleted upon loss of Mll1, which sustains oncogenic Wnt-induced stemness by antagonizing Polycomb Group (PcG)-mediated silencing of key stem cell genes.

## Results

**Mll1 is upregulated in Wnt-high human colon cancer.** We used immunohistochemistry to assess the expression of the histone methyltransferase MLL1 in human colon cancer biopsies. Colon carcinomas showed substantial MLL1 expression at all tumor stages (T0–T4), scored from weak to moderate and strong (Fig. 1a and Supplementary Fig. 1a). The majority of tumors exhibited moderate to strong MLL1 expression, which was not associated to any particular tumor stage. We used a large transcriptomic colon cancer patient data set (n = 1095) to evaluate the association of MLL1 with disease progression. These analyses showed no significant changes in MLL1 expression across tumor stages

(Supplementary Fig. 1b). Patients of stages I-III and stage IV were stratified in three groups based on the level of MLL1 expression (MLL1 low, medium, and high, Supplementary Fig. 1c). High MLL1 levels were associated with shorter survival and increased risk of disease relapse (Supplementary Fig. 1c, d). Immunostaining for β-catenin revealed high levels of nuclear β-catenin, indicative of high Wnt signaling, in a major portion of the analyzed tumors (Supplementary Fig. 1e). Across all tumor stages, carcinomas with high levels of nuclear β-catenin showed strong expression of MLL1 (Fig. 1b and Supplementary Fig. 1f, g). The data reveal that high MLL1 levels in colon cancer are associated with poor patient survival and correlate to high Wnt activity.

**Mll1 is highly expressed in Wnt-activated crypts and Lgr5+ intestinal stem cells.** Lgr5+ intestinal stem cells have been identified as the cells-of-origin in colon cancer[14]. High Wnt activity (Wnt^high) in the intestinal crypts sustains Lgr5+ stem cells that constantly produce secretory and absorptive progenitors. These progenitors migrate out of the Wnt^high niche and differentiate when they reach the villus compartment enriched for differentiation-inducing factors such as Bmp4[15,25] (see schemes in Fig. 1c). We used immunofluorescence to examine the expression of the histone methyltransferase Mll1 along the crypt–villus axis of mice. Mll1 expression is high at the base of the crypt and in Ki67+ proliferating cells of the transit-amplifying (TA) zone. Mll1 is gradually lost towards the differentiated compartment (Fig. 1d and Supplementary Fig. 1h). We observed the highest expression of Mll1 in the stem cells which were identified by the Lgr5-EGFP-IRES-Cre^ERT2 reporter[16] (Fig. 1e, white arrows). Paneth cells were distinguished from adjacent stem cells by immunofluorescence for Mmp7[26] (Fig. 1e, white asterisks). Quantitative immunofluorescence confirmed the strongest expression of Mll1 in Lgr5+ stem cells; it gradually decreased in the TA cells, Paneth cells, and enterocytes (Fig. 1f). A crypt–villus gradient was also observed for H3K4 tri-methylation (H3K4me3) (Supplementary Fig. 1i). We observed a similar distribution of Mll1, Ki67, and H3K4me3 in the mouse colon (Supplementary Fig. 1j–l).

The high expression of Mll1 in the Wnt^high crypt compartment and particularly in Lgr5+ stem cells suggested that Mll1 is crucial for stemness and is lost upon differentiation. The composition of growth factors that are essential for intestinal organoid cultures mimics the intestinal crypt stem cell niches[27]. Modulating stem cell niche properties in small intestinal organoids through the removal of Noggin and administration of Bmp4 led to a reduction of the expression of Mll1, accompanied by a repression of the stem cell gene Lgr5 in a concentration-dependent manner (Fig. 1g). The activation of Bmp signaling and the triggering of differentiation was confirmed by our observation of an increase in the expression of the Bmp target gene Id2[28]. Stimulating Wnt activity by adding the Wnt ligand Wnt3a increased the expression of Mll1 (Fig. 1g). Wnt3a treatment also increased the expression of the Wnt-regulated stem cell gene Lgr5 and the classical Wnt target Axin2[11] (Fig. 1g). Inverse Wnt/Bmp gradients define the crypt stem cell niches and the differentiated villus compartments of the intestinal epithelium[15,25]. Our data thus reveal that the opposing Wnt/Bmp gradients restrict Mll1 expression to the Wnt-activated crypt cells.

**Mll1 is required for the β-catenin^GOF-induced intestinal tumorigenesis.** We addressed the role of Mll1 in Wnt-driven intestinal tumorigenesis using a tamoxifen-inducible genetic mouse model[16,29]. Lgr5-EGFP-IRES-Cre^ERT2; β-catenin^deltaEx3/+ mice produce a stabilized oncogenic form of β-catenin (β-cat^GOF, deleted for the N-terminal phosphorylation and ubiquitination

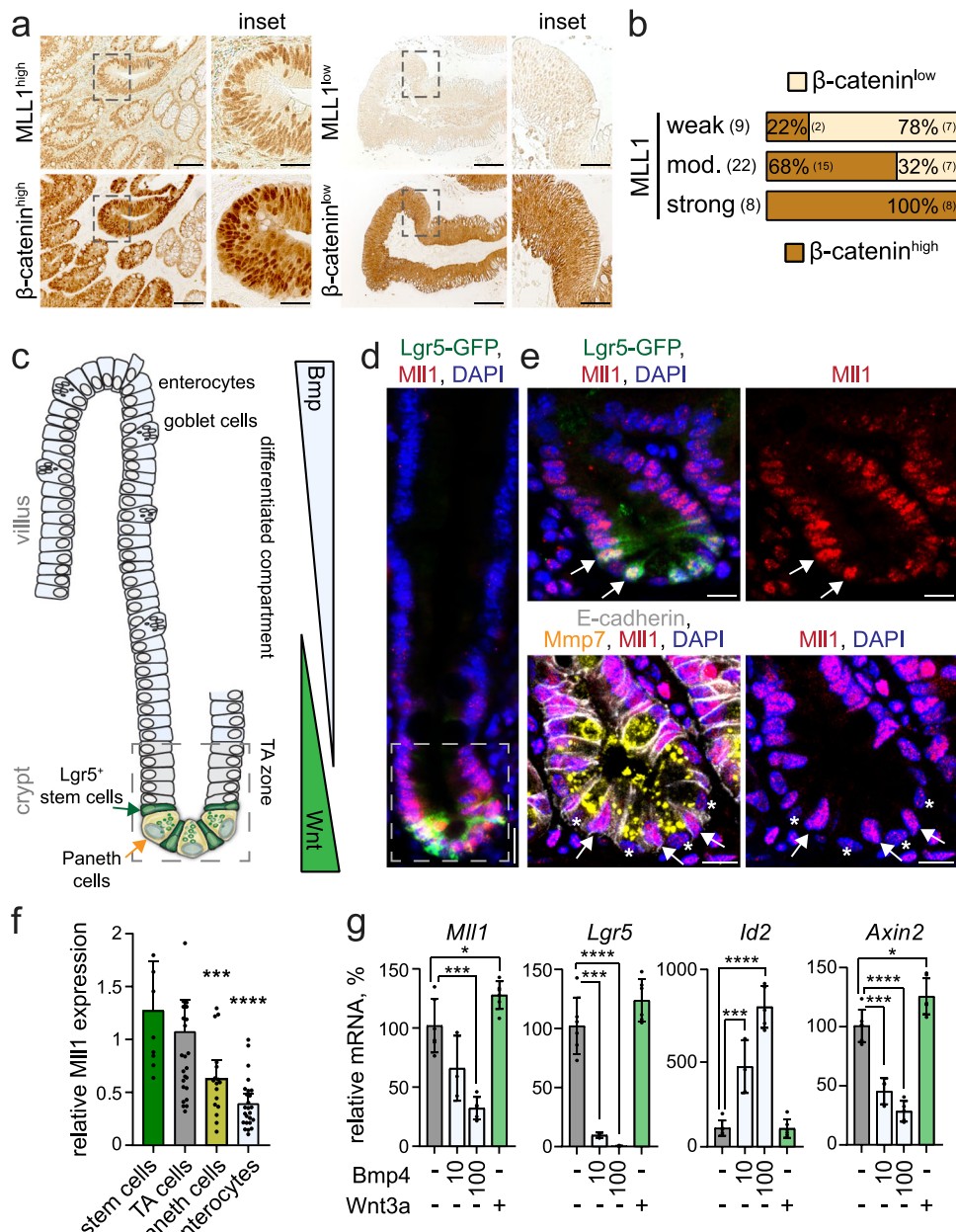

**Fig. 1 High Mll1 expression in colon carcinomas and Lgr5+ intestinal stem cells. a** Immunohistochemistry for MLL1 (upper panel) and β-catenin (lower panel) of colon cancer patient biopsies, tumor stage T0, scale bar 50 μm. Insets on the right, scale bar 20 μm. Stainings were performed on eight independent biopsies. **b** Quantification of nuclear β-catenin staining in tumors with strong, moderate, and weak expression of MLL1, scored as low and high across all tumor stages, $n = 39$ biologically independent colon cancer specimen. Chi-square test for significance, $\chi^2 = 11.5$, $p = 0.0032$. **c** Scheme of the murine small intestine crypt–villus axis[63]. In the lower crypt, Lgr5+ stem cells (green) intersperse between Paneth cells (yellow). Stem cells produce progeny that proliferate and differentiate in the transit-amplifying (TA) zone. Terminally differentiated cells, e.g. enterocytes and goblet cells, move into the villus. Opposing gradients of Wnt and Bmp signaling specify proliferation and differentiation along the crypt–villus axis. **d** Representative immunofluorescence staining for Mll1 (red) on the small intestine of Lgr5-GFP mice reveals a gradual decrease in Mll1 expression from the crypt to villus. Stem cells marked by Lgr5-GFP (green), nuclei in blue (DAPI), scale bar 20 μm. Stainings were performed in five independent mice. **e** Magnification of the crypt region; upper panel, arrows mark stem cells with high Mll1 expression (red); bottom panel, asterisks mark Paneth cells stained by Mmp7 (yellow) with low Mll1 expression. E-cadherin (white) stains cell borders. Nuclei in blue (DAPI), scale bar 10 μm. **f** Quantification of Mll1 staining intensity in nuclei of crypt and villus cells, normalized to the mean Mll1 expression in TA cells located just above the stem cell niche (up to relative + 8 position), quantified from independent stainings of five mice. Data are presented as mean values ± 95% confidence interval, outliers are shown as single dots. Mann–Whitney test (two-tailed) for significance relative to stem cells, ***$p = 0.0009$, ****$p < 0.0001$. **g** Mll1, Lgr5, Id2, and Axin2 mRNA expression in intestinal organoids of mice treated with 10 ng/ml and 100 ng/ml Bmp4 and 0.66 μg/ml recombinant Wnt3a for 24 h, $n = 6$ independent experiments for control and Wnt3a, $n = 3$ for 10 ng/ml Bmp4, $n = 5$ for 100 ng/ml Bmp4, two-tailed unpaired t-test, Mll1 *$p = 0.0329$, ***$p = 0.0001$; Lgr5 ****$p < 0.0001$, ***$p = 0.0004$; Id2 ****$p < 0.0001$, ***$p = 0.0008$; Axin2 *$p = 0.0136$, ****$p < 0.0001$, ***$p = 0.0005$. Data are presented as mean values ± SD. Source data are provided as a Source Data file.

sites) in Lgr5$^+$ stem cells, which is known to induce tumorigenesis. We then crossed the Lgr5-EGFP-IRES-Cre$^{ERT2}$; β-cat$^{GOF}$ mice with Mll1$^{flox}$ mice[30] to ablate Mll1 specifically in the Lgr5$^+$ stem cells. The activation of β-catenin and loss of Mll1 in stem cells was induced in Lgr5-EGFP-IRES-Cre$^{ERT2}$; β-cat$^{GOF}$; Mll1$^{flox/+}$ and Lgr5-EGFP-IRES-Cre$^{ERT2}$; β-cat$^{GOF}$; Mll1$^{flox/flox}$ mice by intraperitoneal injection of tamoxifen (hereafter called β-cat$^{GOF}$; Mll1$^{+/-}$ and β-cat$^{GOF}$; Mll1$^{-/-}$ mice). Mice were analyzed at several time points up to 100 days post-induction. The intestines of the β-cat$^{GOF}$; Mll1$^{+/-}$ mice became severely dysplastic within 30 days and developed adenomas which were highly proliferative, as indicated by Ki67 staining, and contained islets with high Mll1 expression (Fig. 2a, upper panel). In contrast, the homozygous deletion of Mll1 led to the normal crypt–villus structures in β-cat$^{GOF}$; Mll1$^{-/-}$ mice, with proliferation restricted to the crypt regions (Fig. 2a, lower panel). We observed Mll1-depleted crypts among non-recombined Mll1-expressing crypts (Fig. 2a, inset in the lower right image). Thus, the homozygous deletion of Mll1 prevented β-catenin$^{GOF}$-induced intestinal tumorigenesis, as also shown by a Kaplan–Meier plot of tumor incidence (Fig. 2b).

**Mll1 sustains the β-catenin$^{GOF}$-driven tumorigenic Lgr5$^+$ stem cell expansion**. The dysplastic tissue in the β-cat$^{GOF}$; Mll1$^{+/-}$ intestines showed an expansion of Lgr5-GFP$^+$ stem cells (Fig. 2c, upper picture, Supplementary Fig. 2a) and an increase in the stem cell genes *Lgr5*, *Smoc2*, and *Cd44* (Supplementary Fig. 2b). The stem cell expansion in β-cat$^{GOF}$ tumorous intestines was associated with an increase in Mll1 protein levels (Supplementary Fig. 2c, the mutant β-catenin allele is marked by an arrow, quantification is shown below). Specifically, tumor cells with high levels of β-catenin expressed high nuclear levels of Mll1 (Supplementary Fig. 2d; green arrows mark β-catenin$^{high}$ cells, white arrowheads indicate β-catenin$^{low}$ cells). Remarkably, the homozygous loss of Mll1 prevented the Wnt/β-catenin-mediated expansion of stem cells (Fig. 2c, lower picture). A LacZ reporter allele[31] allowed us to trace recombined cells and their progeny. Equal recombination was observed at day 5 in 25% of crypts (Supplementary Fig. 2e, upper row, quantification in f). Subsequently, the mutant cells in β-cat$^{GOF}$; Mll1$^{+/-}$ mice expanded at day 10 and formed LacZ$^+$ dysplasia at day 30 (Supplementary Fig. 2e, middle and lower left pictures, quantification in f). In contrast, β-cat$^{GOF}$; Mll1$^{-/-}$ mice showed a gradual reduction of LacZ$^+$ cells down to 10% over time (Supplementary Fig. 2e, lower right picture, quantification in f). At 30 days, the β-cat$^{GOF}$; Mll1$^{-/-}$ mice exhibited crypts with LacZ$^+$ long-lived Paneth cells interspersed with non-mutant (LacZ$^-$) stem cells (Supplementary Fig. 2g, marked by an arrow). Furthermore, we observed crypts that were normally shaped and depleted of Mll1 but with nuclear β-catenin (β-cat$^{GOF}$) and positive for Lgr5-GFP; they exhibited a slight reduction in proliferation, as seen by a decrease in the incorporation of BrdU (Fig. 2c, lower picture, Supplementary Fig. 2h, i). Apoptosis was not detected in β-cat$^{GOF}$; Mll1$^{-/-}$ crypts (Supplementary Fig. 2j).

Intestinal organoids established from Lgr5-EGFP-IRES-Cre$^{ERT2}$ mice recapitulated the pattern of Mll1 expression observed in control (wildtype) intestines: highest expression is seen in Wnt-dependent Lgr5-GFP$^+$ stem cells (Fig. 2d, e, left panel). Organoids from β-cat$^{GOF}$, β-cat$^{GOF}$; Mll1$^{+/-}$ and β-cat$^{GOF}$; Mll1$^{-/-}$ mice were mutated in culture by in vitro recombination with 4-hydroxy-tamoxifen. All β-cat$^{GOF}$ organoids grew independently of R-spondin1 (Supplementary Fig. 2k). β-cat$^{GOF}$ induced a strong expansion of Lgr5-GFP$^+$ stem cells (Fig. 2d, e, second panel). This expansion was maintained in organoids with heterozygous deletion of Mll1, β-cat$^{GOF}$; Mll1$^{+/-}$

(Fig. 2d, e, third panel). In contrast, homozygous deletion of Mll1, β-cat$^{GOF}$; Mll1$^{-/-}$, prevented the expansion of the stem cell population (Fig. 2d, e, fourth panel). Western blotting for Lgr5-GFP confirmed the stem cell expansion phenotype (Fig. 2f, quantification below). β-cat$^{GOF}$; Mll1$^{-/-}$ organoids exhibited a decrease in the expression of the intestinal stem cell genes *Lgr5*, *Smoc2*, and *Olfm4* (Fig. 2g). The expression of the Wnt-regulated stem cell gene *Ascl2*[32] and the classical Wnt target gene *Axin2*[11] were unaffected by deletion of Mll1 (Fig. 2g). Single β-cat$^{GOF}$; Mll1$^{-/-}$ cells re-formed spheroid-shaped organoids over many passages with high efficiency (Supplementary Fig. 2l), demonstrating the proliferative potential and self-renewal capacity of Mll1-deficient β-cat$^{GOF}$ cells. From the seventh passage on, β-cat$^{GOF}$; Mll1$^{-/-}$ cells formed less organoids, which were smaller and acquired a thick-walled shape (Supplementary Fig. 2m), indicating differentiation and progressive exhaustion of stemness upon loss of Mll1.

Under non-oncogenic conditions without genetic stabilization of β-catenin, mutant Mll1$^{-/-}$ stem cells produced progeny up to 20 days after the induction of mutagenesis, as seen by LacZ lineage tracing in Mll1$^{-/-}$ mice (Supplementary Fig. 3a, quantification on the right). From 20 days onwards, Mll1$^{-/-}$ crypts were progressively lost, but a substantial number of Mll1$^{-/-}$ crypts persisted beyond 50 days after induction (Supplementary Fig. 3b, quantification on the right). The Mll1$^{-/-}$ mice did not show any obvious abnormalities, the intestinal epithelium remained intact, and the overall organ function was not visibly impeded. In summary, these data demonstrate by genetic means that Mll1 is essential for the β-cat$^{GOF}$-dependent stem cell expansion and tumorigenesis. The loss of Mll1 did not interfere with β-cat$^{GOF}$ function per se, but β-cat$^{GOF}$; Mll1$^{-/-}$ stem cells progressively lost their self-renewal capacity. This indicates that Mll1 is crucial to maintain the stemness of β-catenin-activated intestinal stem cells.

**Ablation of Mll1 induces differentiation of β-catenin-activated stem cells**. We isolated the Lgr5-GFP$^+$ intestinal stem cells from β-cat$^{GOF}$; Mll1$^{+/-}$ and β-cat$^{GOF}$; Mll1$^{-/-}$ mice using FACS and analyzed transcriptomic changes by RNA sequencing. This revealed that β-cat$^{GOF}$; Mll1$^{+/-}$ stem cells became Paneth cell-like, which was caused by high Wnt activity due to the β-cat$^{GOF}$ mutation[33]. Immunofluorescence of β-cat$^{GOF}$; Mll1$^{+/-}$ tumor sections and RT-PCR analysis of β-cat$^{GOF}$-mutants revealed a strong expression of the Paneth cell marker Mmp7[26] (Fig. 3a). A volcano plot of the differentially expressed genes in β cat$^{GOF}$; Mll1$^{-/-}$ versus β-cat$^{GOF}$; Mll1$^{+/-}$ stem cells showed both up- and downregulation of genes (Fig. 3b and Supplementary Data 1). This included a global increase in the expression of goblet cell-specific genes, among these *Tff3* (*Itf*) and *Muc2*[34]. Downregulated genes included specific markers of Paneth cells, e.g., *Defa5*, *Defa17*, *Lyz1*[26], indicating that ablation of Mll1 shifted the Paneth-like identity of β-cat$^{GOF}$ stem cells towards a goblet cell fate. Genes specific for enteroendocrine and tuft cells did not undergo major changes (Supplementary Fig. 3c). β-cat$^{GOF}$; Mll1$^{-/-}$ stem cells exhibited a strongly increased expression of *Atoh1*, a transcription factor essential for secretory lineage determination[35], and its target gene *Spdef*, which instructs secretory cell differentiation in Paneth and goblet cells[36,37] (Fig. 3b). The expression of *Hes1*, a Notch-controlled transcription factor that represses *Atoh1* expression[38], was not downregulated in the β-cat$^{GOF}$; Mll1$^{-/-}$ stem cells. There were also no decreases in other Notch target genes such as *Hes5* and *Hey1* (Supplementary Data 1), which means that the upregulation of *Atoh1* occurs independently of a global decrease in Notch signaling. The Paneth cell gene *Mmp7* was increased in β-cat$^{GOF}$; Mll1$^{-/-}$ stem cells, indicating priming

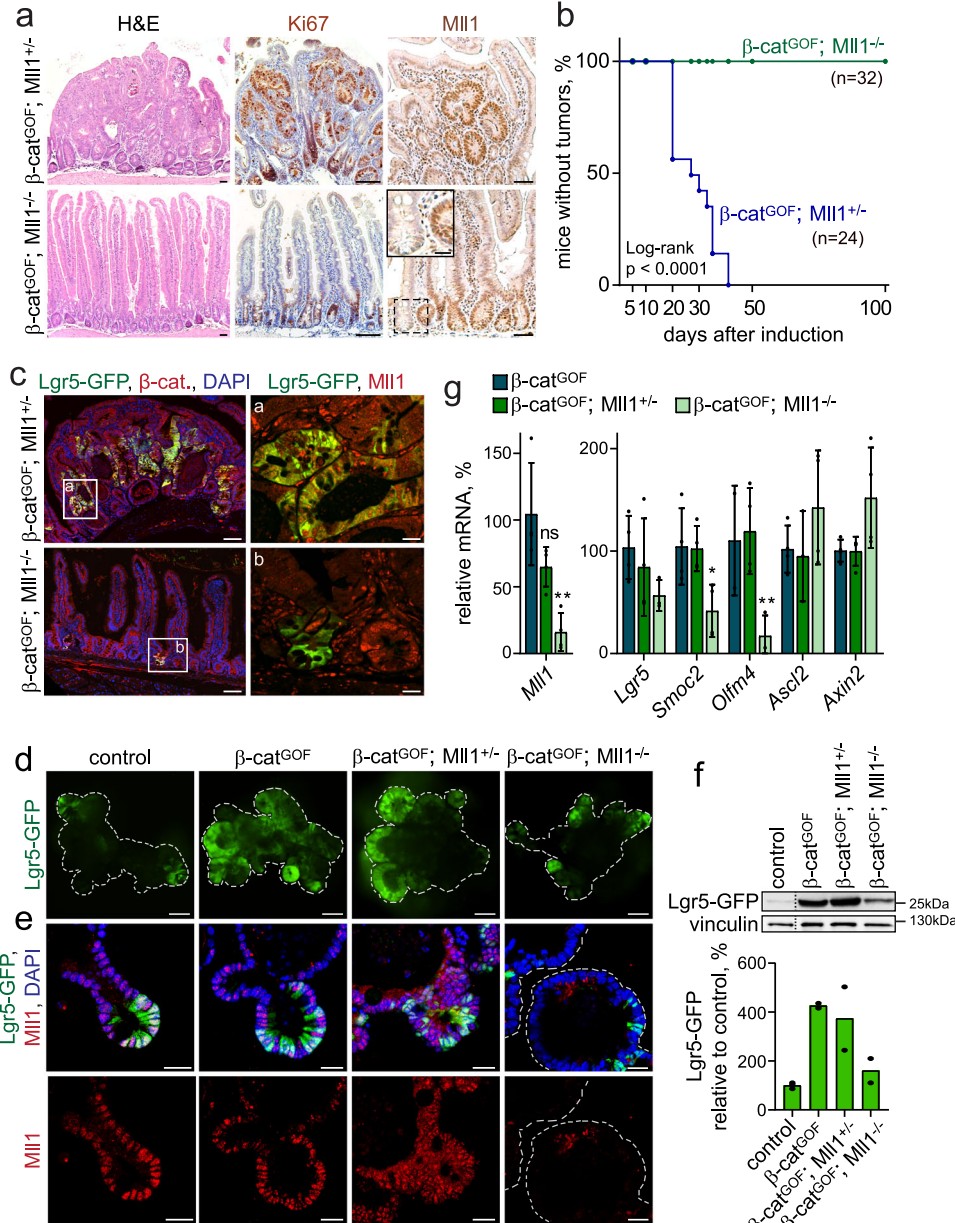

**Fig. 2 The β-catenin$^{GOF}$-induced intestinal stem cell expansion and tumorigenesis depend on Mll1. a** Sections of intestines of Lgr5-EGFP-IRES-Cre$^{ERT2}$; β-cat$^{GOF}$; Mll1$^{+/−}$ and Lgr5-EGFP-IRES-Cre$^{ERT2}$; β-cat$^{GOF}$; Mll1$^{−/−}$ mice, called β-cat$^{GOF}$; Mll1$^{+/−}$ and β-cat$^{GOF}$; Mll1$^{−/−}$, at 30 days after induction with tamoxifen: H&E staining (left), scale bar 50 μm; Ki67 staining (middle), nuclei counterstained with hematoxylin, scale bar 100 μm; Mll1 immunohistochemistry (right), scale bar 50 μm, with the magnification of Mll1 knockout crypt in inset, scale bar 20 μm. Stainings were performed on sections of three independent mice per genotype. **b** Kaplan–Meier plot of tumor incidence in β-cat$^{GOF}$; Mll1$^{+/−}$ and β-cat$^{GOF}$; Mll1$^{−/−}$ mice. **c** Immunofluorescence for Lgr5-GFP stem cells (green) and β-catenin (red) on intestinal sections of β-cat$^{GOF}$; Mll1$^{+/−}$ and β-cat$^{GOF}$; Mll1$^{−/−}$ mice at day 30 after induction, scale bar 100 μm. Insets **a**, **b** immunostaining for Lgr5-GFP (green) and Mll1 (red) on serial sections, scale bar 20 μm. Stainings were performed on sections of three independent mice. **d** Fluorescence live imaging of non-induced (control) and tamoxifen-induced Lgr5-GFP-IRES-Cre$^{ERT2}$; β-cat$^{GOF}$, Lgr5-GFP-IRES-Cre$^{ERT2}$; β-cat$^{GOF}$; Mll1$^{+/−}$ and Lgr5-GFP-IRES-Cre$^{ERT2}$; β-cat$^{GOF}$; Mll1$^{−/−}$ mouse intestinal organoids. Lgr5-GFP (green) marks intestinal stem cells, scale bars 100 μm. Representative images from three independent experiments with three biologically independent organoid lines. **e** Immunofluorescence staining for Lgr5-GFP (green) and Mll1 (red) on sections of organoids of the four genotypes, nuclei in blue (DAPI), scale bars 30 μm. Representative stainings from three independent experiments. **f** Western blot for Lgr5-GFP in control and tamoxifen-induced β-cat$^{GOF}$, β-cat$^{GOF}$; Mll1$^{+/−}$, and β-cat$^{GOF}$; Mll1$^{−/−}$ organoids of mice, vinculin as a control for equal loading. Quantification of Lgr5-GFP protein levels in mutant organoids relative to controls below, $n = 2$ samples from two independent experiments processed in parallel for western blotting. **g** mRNA expression of *Mll1* and intestinal stem cell genes in β-cat$^{GOF}$; Mll1$^{+/−}$ and β-cat$^{GOF}$; Mll1$^{−/−}$ mouse organoids relative to β-cat$^{GOF}$ organoids, $n = 4$ independent experiments, two-tailed unpaired t-test, *Mll1* **$p = 0.005$, *Smoc2* *$p = 0.032$, *Olfm4* **$p = 0.037$. Data are presented as mean values ± SD. Source data are provided as a Source Data file.

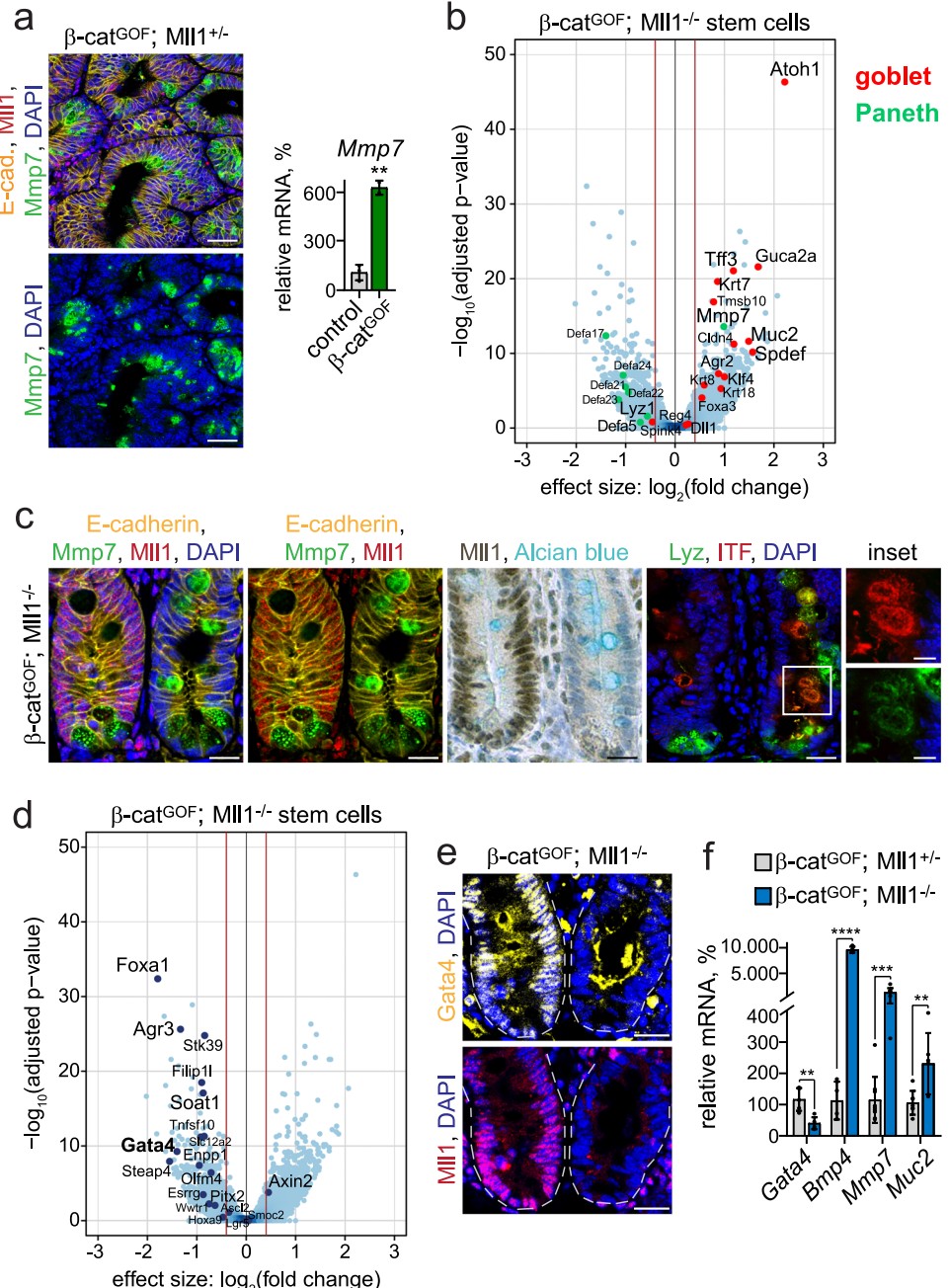

**Fig. 3 The ablation of Mll1 induces differentiation. a** Left: Immunostaining for Mmp7 (green) and Mll1 (red) on β-cat$^{GOF}$; Mll1$^{+/-}$ tumor sections, scale bars 50 μm. Nuclei stained with DAPI (blue), E-cadherin (yellow) stains cell borders. Right: mRNA expression of *Mmp7* in control and β-cat$^{GOF}$ intestines, $n = 3$ independent mice, two-tailed unpaired *t*-test, **$p = 0.0012$. Data are presented as mean values ± SD. **b** Volcano plot of differentially expressed genes in β-cat$^{GOF}$; Mll1$^{-/-}$ relative to β-cat$^{GOF}$; Mll1$^{+/-}$ Lgr5-GFP$^+$ stem cells isolated from four independent mice each, represented as blue dots. Cut-off log$_2$ fold-change ≥ 0.5. Goblet and Paneth cell-specific genes[78] are marked in red and green, respectively, and indicated by names. **c** Immunostaining on sections of adjacent non-recombined (left) and recombined (right) small intestinal crypts in β-cat$^{GOF}$; Mll1$^{-/-}$ mice, left: for Mmp7 (green) and Mll1 (red), middle: for Mll1 (brown) and Alcian blue; nuclear counterstaining with hematoxylin. Right: immunostaining for Lyz (green) and ITF (red), scale bar 20 μm. Insets on the right: magnifications of mutant double-positive cells, scale bars 10 μm. Nuclei stained with DAPI (blue), E-cadherin (yellow) stains cell borders. Stainings were performed in two independent mice per genotype. **d** Volcano plot of differentially expressed stem cell genes and transcription factors in β-cat$^{GOF}$; Mll1$^{-/-}$ relative to β-cat$^{GOF}$; Mll1$^{+/-}$ Lgr5-GFP$^+$ stem cells isolated from four independent mice each, represented as dark blue dots. Cut-off log$_2$ fold-change ≥ 0.5. **e** Immunostaining for Gata4 (yellow, upper panel) and Mll1 (red, lower panel) on sections of adjacent non-recombined (left) and recombined (right) small intestinal crypts in β-cat$^{GOF}$; Mll1$^{-/-}$ mice, scale bars 20 μm. Nuclei stained with DAPI (blue). Stainings were performed on sections of six independent mice. **f** mRNA expression of *Gata4*, *Bmp4* ($n = 4$ independent experiments) and the secretory differentiation markers *Mmp7* and *Muc2* ($n = 7$ independent experiments) in β-cat$^{GOF}$; Mll1$^{+/-}$ and β-cat$^{GOF}$; Mll1$^{-/-}$ organoids, two-tailed unpaired *t*-test, *Gata4* **$p = 0.0032$, *Bmp4* ****$p < 0.0001$, *Mmp7* ***$p = 0.001$, *Muc2* **$p = 0.0096$. Data are presented as mean values ± SD. Source data are provided as a Source Data file.

towards a mixed Paneth-goblet state. Mll1-deficient crypts of β-cat$^{GOF}$; Mll1$^{-/-}$ mutant mice exhibited large cells that were double-positive for Paneth cell (Mmp7 and Lyz) and goblet cell markers (Alcian blue (mucin) and ITF) (right crypt in Fig. 3c). Such Paneth-goblet intermediate cells were not observed in adjacent non-recombined Mll1-expressing crypts on the same section (left crypt in Fig. 3c).

RNA sequencing of the sorted Lgr5-GFP$^+$ stem cells also revealed that β-cat$^{GOF}$; Mll1$^{-/-}$ stem cells exhibited a decreased expression of several transcription factors and stem cell genes. As illustrated in a volcano plot, Mll1-deficient β-cat$^{GOF}$ stem cells showed a decreased expression of the intestinal stem cell gene *Olfm4*. *Lgr5* and *Smoc2* expression was not significantly altered (Fig. 3d). The ablation of Mll1 did not affect the expression of the classical Wnt target gene *Axin2* (Fig. 3d), as was observed in β-cat$^{GOF}$; Mll1$^{-/-}$ organoids. In β-cat$^{GOF}$; Mll1$^{-/-}$ stem cells, the expression of the transcription factors Foxa1 and Gata4 was downregulated (Fig. 3d), the latter of which had previously been proposed as an Mll1 target gene[39]. Immunostaining confirmed a strong reduction of the GATA binding protein 4 (Gata4) in Mll1-deficient crypts, compared to non-recombined Mll1-expressing crypts on sections of β-cat$^{GOF}$; Mll1$^{-/-}$ small intestines (Fig. 3e). Gata4/6 transcription factors had previously been reported to repress *Bmp4*, thereby maintaining colon cancer stemness[40]. RT-PCR analysis of β-cat$^{GOF}$; Mll1$^{-/-}$ organoids confirmed a decreased expression of *Gata4* and revealed a concomitant upregulation of *Bmp4* and the secretory differentiation markers *Mmp7* and *Muc2* (Fig. 3f), identifying Mll1 as an upstream regulator of Gata4/Bmp4-controlled cancer stemness and differentiation. In conclusion, the stem cell transcriptome showed that ablation of Mll1 primed β-cat$^{GOF}$ intestinal stem cells for differentiation and shifted the Wnt-imposed Paneth-like identity towards a goblet cell fate. Thus, Mll1 maintains stemness by restricting the differentiation of β-cat$^{GOF}$ intestinal stem cells.

**Mll1 sustains the stemness and tumorigenicity of human colon cancer cells.** To investigate the essential role of Mll1 in the stemness of human colon tumors, we established stable, doxycycline-inducible shMLL1 single-cell clones of the human colon adenocarcinoma cell lines Ls174T and DLD1, which exhibited a strong reduction of MLL1 (Supplementary Fig. 4a–d). In all experiments, shMLL1 cells were compared with non-induced and non-targeted shRNA (control) cells. The ablation of MLL1 did not affect the expression of other MLL family members (Supplementary Fig. 4e). Serial re-plating assays revealed that the ablation of MLL1 did not immediately affect the viability and proliferation of Ls174T and DLD1 cells in adherent 2D cultures. A significant growth reduction was detected only after four passages and a total of 10–14 days of culture (Supplementary Fig. 4f, g). We found that MLL1 protein was strongly increased in 3D (non-adherent, self-renewing) sphere cultures of human Ls174T cells compared to adherent cultures, while the expression of the family member SETD1A was not changed (Supplementary Fig. 4h, quantification on the right). In contrast to the 2D culture, the ablation of MLL1 in 3D sphere cultures had a rapid effect on the growth and self-renewal of the tumor cells: secondary sphere formation of shMLL1 cells was reduced by 80% in Ls174T and by 60% in DLD1 cells, compared to control cells (Fig. 4a and Supplementary Fig. 4i, quantifications on the right). Residual shMLL1 spheres exhibited minimal MLL1 expression and reduced proliferation, as shown by staining for Ki67 (Supplementary Fig. 4j). shMLL1 spheres were negative for the apoptosis marker cleaved Caspase-3 (Supplementary Fig. 4k, western blot on the right).

To test the in vivo growth and tumor-forming capacities of the shMLL1 cells, doxycycline-induced shMLL1 and control Ls174T colon cancer cells were subcutaneously engrafted in nude mice. Mice inoculated with shMLL1 cells were administered doxycycline throughout the experiment. Control cells formed large tumors within 28 days (Fig. 4b). In contrast, tumor formation by shMLL1 cells was reduced by 75%. The residual xenografts showed a strong reduction of MLL1 (Supplementary Fig. 4l, quantification below) and H3K4me3 (Supplementary Fig. 4m, upper panel), and significantly fewer proliferating cells (Supplementary Fig. 4m, middle panel, quantifications on the right). The apoptosis marker cleaved Caspase-3 did not increase in shMLL1 Ls174T tumors (Supplementary Fig. 4m, lower panel, western blot on the right). The shMLL1 human Ls174T xenografts and sphere cells showed enhanced differentiation with an increase in the expression of the Paneth cell-specific genes *DEFA5* and *LYZ* and the goblet cell markers *MUC2* and *ITF* (Supplementary Fig. 4n), as was also observed in the β-cat$^{GOF}$ mouse tumor model. An increase in *ITF* expression was also observed in DLD1 sphere cells (Supplementary Fig. 4o). The expression of the differentiation-associated cell cycle inhibitor *p21* was upregulated in shMLL1 Ls174T xenografts and spheres (Supplementary Fig. 4p, q, quantifications on the right). These data show that MLL1 is crucial for stemness and tumor formation of human colon cancer cells.

**Mll1 regulates specific stem cell genes to sustain colon cancer stemness.** We used RNA sequencing to investigate transcriptomic changes upon knockdown of MLL1 in the human colon cancer cells. Comparison of the differentially expressed genes with a signature of Mll1-ablated fibroblasts[39] revealed deregulation of MLL1 target genes such as *SMOC2* and *ENPP1* (Supplementary Fig. 4r, Supplementary Data 2). The human shMLL1 colon cancer cells showed a decrease of the expression of *FOXA1* and the GATA4 colon homolog *GATA6*, which we had identified as Mll1-regulated genes in murine β-cat$^{GOF}$; Mll1$^{-/-}$ stem cells. Comparisons with intestinal stem cell signatures reported by the groups of Clevers[41], Soshnikova[42], and Batlle[20] revealed a significant gene overlap (Fig. 4c: Clevers 36, Soshnikova 106, and Batlle 19 genes). We compiled 132 deregulated stem cell genes to establish an MLL1-regulated colon cancer stem cell signature (Supplementary Data 3). A volcano plot of the deregulated genes overlaid with this signature showed a predominant down-regulation of stem cell genes upon knockdown of MLL1; these included *OLFM4*, *SMOC2*, *IGFBP4*, and the Wnt-regulated gene *LGR5*[41] (Supplementary Data 3 and Fig. 4d). RT-PCR analyses of sphere cultures confirmed the decrease in the expression of the stem cell genes *LGR5*, *IGFBP4*, *SMOC2*, and *OLFM4*. The expression of *CD44* and *LRIG1*[41] was not significantly decreased. There was no reduction in the expression of the Wnt-regulated genes *ASCL2*, *SOX9*[43], and *AXIN2* (Fig. 4e). The selective regulation of Wnt-dependent stem cell genes was also observed in DLD1 human colon cancer cells (Supplementary Fig. 4s). The ablation of MLL1 did not affect the overall activity of Wnt signaling, as shown by a Tcf/Lef luciferase reporter assay (TOPflash)[44] in comparison to cells either treated with the Wnt activator CHIR99021[45] or the Wnt inhibitor LF3[46] (Fig. 4f).

**Mll1 regulates the H3K4me3–H3K27me3 balance at specific stem cell genes.** Chromatin immunoprecipitation (ChIP) revealed high H3K4me3 and low H3K27me3 levels at the transcription start sites (TSS) of the MLL1-regulated stem cell genes *LGR5*, *SMOC2*, *OLFM4*, and *IGFBP4* (Fig. 5a and Supplementary Fig. 5a). Ablation of MLL1 shifted the epigenetic state of the stem

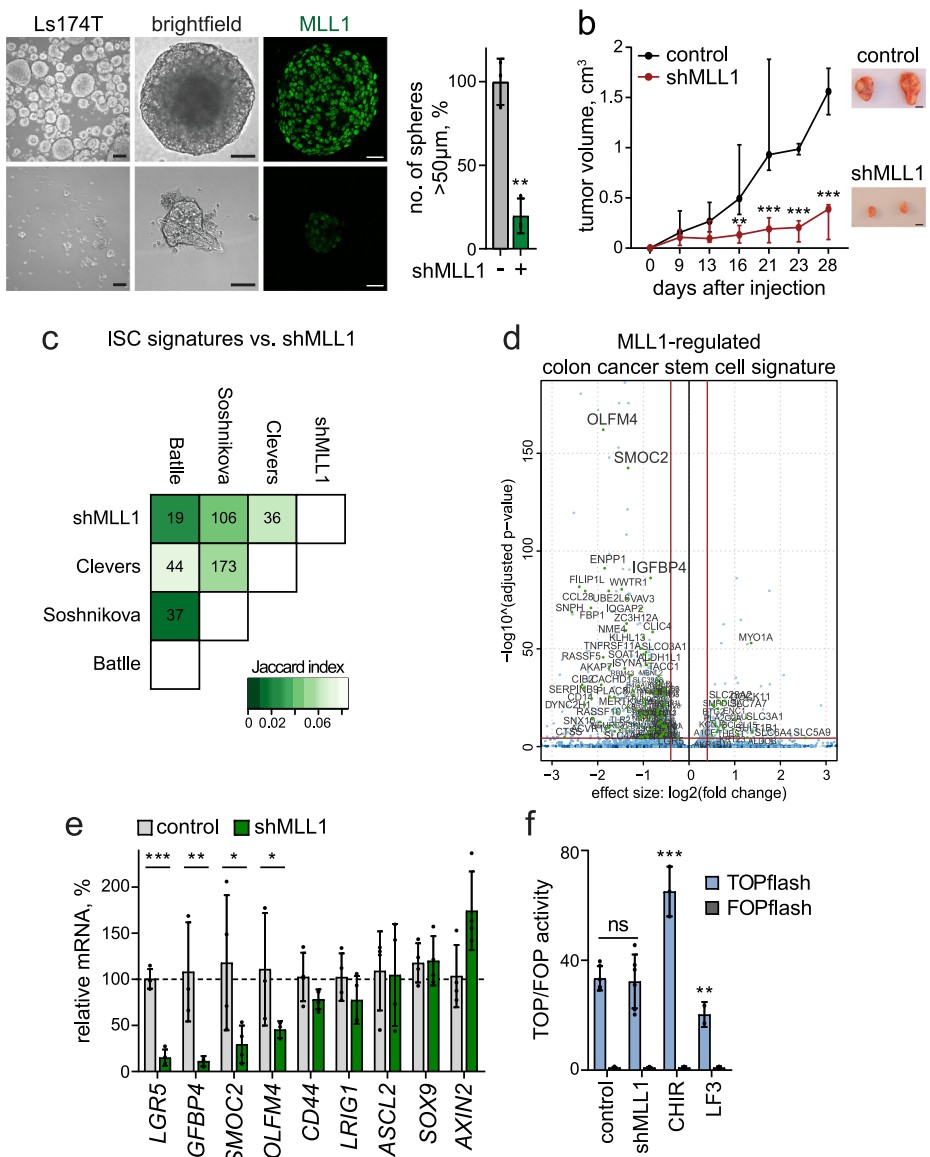

**Fig. 4 Mll1 sustains colon cancer stemness. a** Brightfield images and MLL1 immunofluorescence (green) of control and 6d doxycycline-induced shMLL1 human Ls174T spheres, scale bar 100 μm, magnification of brightfield images 25 μm, immunofluorescence 20 μm. Quantification of spheres >50 μm in size on the right, $n = 3$ biologically independent samples, two-tailed unpaired $t$-test, **$p = 0.0013$. Data are presented as mean values ± SD. **b** Xenografts of control and shMLL1 Ls174T cells in NMRI[nu/nu] mice. Mice inoculated with shMLL1 cells were administered doxycycline from day −1 throughout the experiment. Left: tumor growth curves plotted as median with range, $n = 6$ independent animals inoculated with three biologically independent induced or non-induced shMLL1 Ls174T cell clones in duplicates over two independent experiments, two-tailed unpaired $t$-test, d16 **$p = 0.007$, d21 ***$p = 0.0034$, d23 ***$p = 0.00016$, d28 ***$p = 0.00018$. Right: dissected tumors at day 28, scale bar 2 cm. **c** Differentially expressed genes in shMLL1 human Ls174T cells, cut-offs $\log_2$ fold-change ≥ 0.5, adjusted $p$-value ≤ 0.05, compared to intestinal stem cell signatures reported by the groups of Clevers, Soshnikova, and Batlle. The correlation table indicates the number of common genes, the Jaccard index of similarity is encoded in green color shadings (lighter means more similar). **d** Volcano plot of differentially expressed genes in shMLL1 human Ls174T cells (blue dots), overlaid with MLL1-regulated colon cancer stem cell signature (Supplementary Data 3), marked as green dots and indicated by name. **e** mRNA expression of intestinal stem cell genes and classical Wnt targets in control and 6d doxycycline-induced shMLL1 human Ls174T spheres, $n = 4$ samples derived from three biologically independent cell clones (IGFBP4, OLFM4 $n = 3$), two-tailed unpaired $t$-test, LGR5 ***$p = 0.001$, IGFBP4 **$p = 0.0018$, SMOC2 *$p = 0.015$, OLFM4 *$p = 0.03$. Data are presented as mean values ± SD. **f** Tcf/Lef luciferase reporter assays in control and shMLL1 cells treated with doxycycline for 10 days, $n = 6$ samples over three independent experiments, two-tailed unpaired $t$-test. Wnt-responsive luciferase activity (TOPflash) calculated relative to control (FOPflash). 3 μM CHIR99021 and 50 μM LF3 were added for 24 h to stimulate and repress Wnt signaling, respectively, $n = 3$ independent experiments, two-tailed unpaired $t$-test, ***$p = 0.0002$, **$p = 0.0074$. Data are presented as mean values ± SD. Source data are provided as a Source Data file.

cell genes from transcriptionally permissive (H3K4me3) to repressed (H3K27me3) (Fig. 5a and Supplementary Fig. 5a). There were no changes in levels of H3K4me3 or H3K27me3 at the TSS of ASCL2 and AXIN2 (Fig. 5a, right). Global levels of H3K4 and H3K27 methylation were not changed (Supplementary

Fig. 5b). We identified MLL1 binding to the promoters of the intestinal stem cell genes LGR5, SMOC2, IGFBP4, and OLFM4 (Fig. 5b and Supplementary Fig. 5c, d). ChIP specificity was confirmed by MLL1 binding to MECOM, an MLL1 target gene[47], and by comparison to a negative control region (TAL1 + 70[48]).

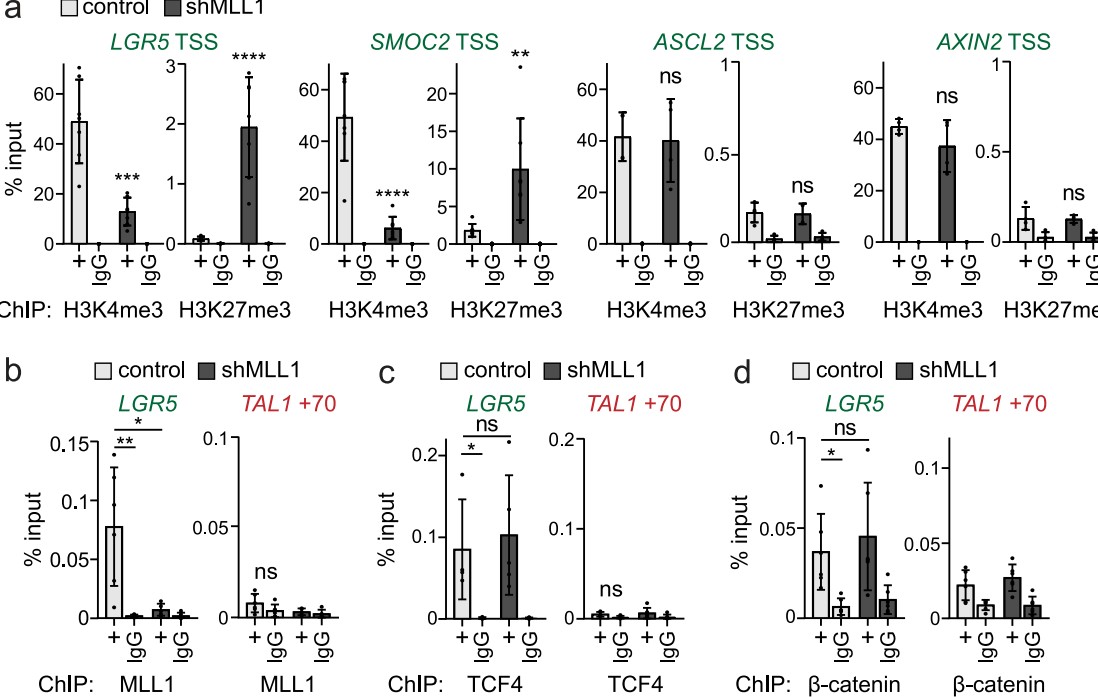

**Fig. 5 Mll1 directly regulates intestinal stem cell genes. a** ChIP for H3K4me3 and H3K27me3 at TSSs of *LGR5*, *SMOC2* (*n* = 7 samples derived from two biologically independent cell clones over four independent experiments), *ASCL2* and *AXIN2* (*n* = 4 samples derived from two biologically independent cell clones over two independent experiments) in non-induced control (light gray columns), and 11d doxycycline-induced shMLL1 Ls174T cells (black columns), represented as % input. Two-tailed unpaired *t*-test, significance calculated for control versus shMLL1, *LGR5* TSS ***p = 0.0002, ****p < 0.0001; *SMOC2* TSS ****p < 0.0001, **p = 0.0082. **b** ChIP for MLL1 in non-induced control (light gray columns) and 6d doxycycline-induced shMLL1 human Ls174T cells (black columns) at the *LGR5* promoter, negative control region + 70 kb downstream of the *TAL1* promoter, represented as % input, *n* = 6 control and *n* = 5 shMLL1 samples derived from two biologically independent cell clones over four independent experiments. Two-tailed unpaired *t*-test, significance calculated for control versus shMLL1 and IgG, *p = 0.0129, **p = 0.0042. ChIP for TCF4 (**c**) and β-catenin (**d**) at the *LGR5* promoter in non-induced control (light gray columns) and 6d doxycycline-induced shMLL1 human Ls174T cells (black columns), negative control region *TAL1* + 70, represented as % input, *n* = 5 (TCF4) and *n* = 6 samples (β-catenin) derived from two biologically independent cell clones over three independent experiments. Two-tailed unpaired *t*-test, significance calculated for control versus shMLL1 and IgG, TCF4 *p = 0.03, β-catenin *p = 0.02. All data are presented as mean values ± SD. Source data are provided as a Source Data file.

Strongly reduced binding of MLL1 was observed in shMLL1 Ls174T and DLD1 cells (Fig. 5b and Supplementary Fig. 5c–e). At the *LGR5* promoter, ChIP revealed no change in β-catenin/TCF4 occupancy upon MLL1 ablation (Fig. 5c, d). In the absence of MLL1, *LGR5* expression is thus repressed by promoter-associated H3K27me3, despite occupancy of the promoter by β-catenin/TCF4. Our data indicate that MLL1 controls stem cell gene activity by modulating active and repressive promoter histone methylation marks.

**Antagonistic control of *Lgr5* by Mll1 and PcG.** H3K27me3 marks are deposited by the polycomb repressive complex 2 (PRC2)[24]. Inhibition of the catalytic PRC2 subunit EZH2 with Gsk126[49] partially restored the decreased expression of the MLL1-regulated stem cell genes *LGR5* and *OLFM4*, even in the absence of MLL1 (Fig. 6a). H3K4me3 marks at the TSS of *LGR5* and *OLFM4* were not restored upon Gsk126 treatment (Fig. 6b), indicating that the re-initiation of *LGR5* and *OLFM4* expression was independent of MLL1 and its methyltransferase activity. The expression of the MLL1-independent Wnt target genes *ASCL2* and *AXIN2* was unaffected by Gsk126 treatment (Fig. 6c), and H3K4me3 and H3K27me3 TSS marks were unchanged (Fig. 6d). Taken together, our data show that the histone methyltransferase MLL1 sustains the oncogenic Wnt-driven expression of specific stem cell genes by antagonizing

PcG-mediated gene silencing. In the absence of MLL1, PRC2 catalyzes the tri-methylation of H3K27me3 and mediates gene repression. In the case of *LGR5*, this occurs regardless of the promoter occupancy of β-catenin/TCF4 (summarized in the schemes in Fig. 6e). Overall, our data show that Wnt-dependent stem cell genes are differentially controlled by epigenetic regulators, and reveal MLL1 as crucial for the maintenance of human colon cancer stemness.

**Discussion**

Here we demonstrate a crucial role for the epigenetic regulator Mll1 in Wnt/β-catenin-driven intestinal tumor formation and cancer stemness. We show that Mll1 promotes the β-catenin^GOF-induced expansion of Lgr5+ intestinal stem cells. Mll1 is highly expressed in Lgr5+ stem cells and human colon cancer specimens with high levels of nuclear β-catenin. The expression of Mll1 increased upon enhanced Wnt/β-catenin activity, indicating that canonical Wnt signaling controls *Mll1* expression. Recently, we have shown that the Mll1 promoter harbors functional Tcf4 binding sites, and that Mll1 expression is increased in Wnt-driven salivary gland and head and neck tumor models[23]. We now demonstrate that Mll1 mediates a differential response of Wnt-dependent stem cell genes in colon cancer. Mll1 does not interfere with the global activity of Wnt signaling, but controls its outcome at the level of gene regulation.

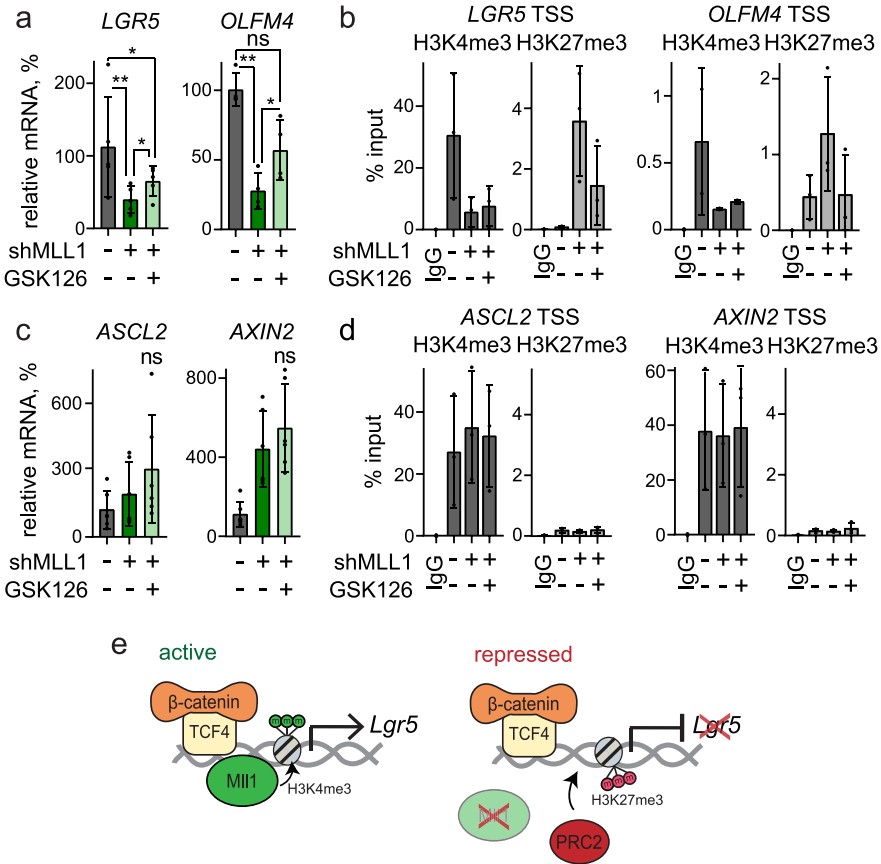

**Fig. 6 Mll1 antagonizes PcG-mediated silencing of stem cell genes. a** mRNA expression of *LGR5* and *OLFM4* in control (dark gray), 6d doxycycline-induced shMLL1 human Ls174T cells (dark green), and shMLL1 Ls174T cells following 4-day Gsk126 treatment (light green), *n* = 4 independent experiments, two-tailed unpaired *t*-test, *LGR5* \**p* = 0.014, \*\**p* = 0.0016, \**p* = 0.027; *OLFM4* \*\**p* = 0.008, \**p* = 0.03. Data are presented as mean values ± SD. **b** ChIP for H3K4me3 (dark gray columns) and H3K27me3 (light gray columns) at the TSSs of *LGR5* and *OLFM4* (*n* = 3 independent experiments) in control, 11d doxycycline-induced shMLL1 human Ls174T cells, and shMLL1 Ls174T cells following 4-day Gsk126 treatment, represented as % input. Data are presented as mean values ± SD. **c** mRNA expression of *ASCL2* and *AXIN2* in control (dark gray), 6d doxycycline-induced shMLL1 human Ls174T cells (dark green), and shMLL1 Ls174T cells following 4-day Gsk126 treatment (light green), *n* = 6 independent experiments, two-tailed unpaired *t*-test. Data are presented as mean values ± SD. **d** ChIP for H3K4me3 (dark gray columns) and H3K27me3 (light gray columns) at the TSSs of *ASCL2* and *AXIN2* (*n* = 3 independent experiments) in control, 11d doxycycline-induced shMLL1 human Ls174T cells, and shMLL1 Ls174T cells following 4-day Gsk126 treatment, represented as % input. Data are presented as mean values ± SD. **e** Scheme of β-catenin/TCF4/MLL1-dependent regulation of *LGR5* in colon cancer cells. MLL1 prevents PRC2-mediated gene silencing to sustain *LGR5* expression. This is associated with the MLL1-dependent H3K4 tri-methylation at the *LGR5* TSS. Ablation of MLL1 results in PRC2-mediated H3K27me3 and silencing of *LGR5* expression. Source data are provided as a Source Data file.

The expression of a stabilized oncogenic form of β-catenin in Lgr5[+] stem cells produced intestinal tumors with a high expression of intestinal stem cell genes. Stem-like gene expression profiles are indicative of high tumor-initiating potential and self-renewal capacity of colon cancer cells[20]. In particular, tumors with increased levels of Lgr5 have malignant potential and are invasive and aggressive[50]. Genetic ablation of Lgr5[+] stem cells by diptheria toxin halts colon cancer growth, revealing the relevance of these stem cells for tumorigenesis[51]. The genetic loss of Mll1 prevented β-catenin[GOF]-driven intestinal tumorigenesis. Ablation of Mll1 halted tumor growth in a xenograft model and reduced the Wnt-imposed stem cell gene expression profile in colon cancer cells. We found that Mll1 controls selective stem cell genes. It regulated the expression of the stem cell marker Lgr5[16], whereas it did not regulate the expression of the Wnt targets Axin2[11], Sox9[43], or Ascl2[32]. Lgr5 expression critically depends on Mll1: in the absence of Mll1, β-catenin binding was not sufficient to activate Lgr5 expression. Mll1 also controls the intestinal stem cell genes Smoc2, Igfbp4, and Olfm4, which are induced in Wnt-mutated cancer cells but regulated by other signaling pathways such as Notch and NF-κB[52,53].

Mechanistically, Mll1 sustained the expression of stem cell genes in Wnt-activated tumor cells by antagonizing PcG-mediated silencing. Mll1 knockdown resulted in a switch of activating H3K4me3 to repressive H3K27me3 chromatin marks at the transcription start sites of Lgr5, Smoc2, Igfbp4, and Olfm4. Strikingly, the inhibition of the polycomb H3K27 tri-methyltransferase Ezh2 with Gsk126[54] partially restored the expression of Lgr5 in Mll1-deficient cells despite the absence of Mll1 and H3K4me3 marks. This implies that Mll1 and its H3K4me3 activity are dispensable for initiating Lgr5 expression but crucial for preventing PRC2-mediated silencing of the Lgr5 promoter[55]. These observations concord with the opposition between trithorax and Polycomb group proteins in *Drosophila*[56]. Remarkably, the genetic inactivation of the polycomb complex PRC1 in oncogenic Wnt-activated intestinal stem cells also has a tumor-preventive effect[57], implying that PRC1 plays a role in promoting Wnt-driven intestinal tumorigenesis. Further, PRC2 ablation in the mouse intestine causes loss of stem cells and aberrant secretory differentiation by deregulating Wnt signaling and cell proliferation[58]. Apparently, a fine-tuned balance of active and repressive histone modifications is essential for the

maintenance of intestinal stem and tumor-initiating cells, a regulatory mechanism of stemness in which Mll1 is crucially involved.

Lineage tracing experiments revealed that the number of Mll1-deficient cells diminished over time. This was not due to apoptosis caused by the ablation of Mll1. Individual Mll1-deficient intestinal stem cells in β-cat[GOF]; Mll1[−/−] mice persisted up to 100 days after induction and were capable of producing progeny. These mutant cells did not expand into tumors, although they exhibited nuclear β-catenin. Our studies in organoids revealed that the ablation of Mll1 did not immediately eliminate β-cat[GOF] stem cells but progressively exhausted their stemness. Through transcriptome profiling of FACS-isolated Lgr5[+] intestinal stem cells at 10 days after mutagenesis, we determined that the ablation of Mll1 in β-cat[GOF] stem cells instructed a goblet cell differentiation program. Residual Mll1 knockout crypts in β-cat[GOF]; Mll1[−/−] mice revealed cells positive for goblet and Paneth cell markers, demonstrating a secretory differentiation of β-cat[GOF] stem cells upon ablation of Mll1. Mll1-deficient stem cells and crypts showed a strong decrease in levels of the transcription factors Foxa1 and Gata4. High levels of Foxa1 expression were observed in poorly differentiated colon cancer, and Gata6, the colon homolog of Gata4, participates in the regulation of stemness by preventing Bmp4-induced differentiation in Wnt-driven colon cancer[40,59]. Moreover, β-cat[GOF]; Mll1[−/−] stem cells exhibited an increase in the expression of *Atoh1*, a transcription factor that instructs secretory lineage differentiation[35]. The ablation of Gata4 has previously been shown to increase intestinal *Atoh1* levels[60]. Gata4-deficient intestines exhibit enhanced levels of Atoh1 and increased numbers of goblet cells[60,61]. Our data support a Gata4-mediated restriction of *Atoh1* expression and goblet cell differentiation. We further identify Mll1 as an upstream regulator of a Gata4–Atoh1 axis that controls stemness versus secretory goblet cell differentiation.

Altogether, our data show that the epigenetic regulator Mll1 promotes the initiation and growth of Wnt-driven colon cancer by maintaining stemness and preventing the differentiation of Wnt-activated tumor-initiating and cancer stem cells. By sustaining Gata4/6 expression and directly controlling specific stem cell genes, Mll1 promotes stemness and prevents secretory goblet cell differentiation of Wnt-activated cancer cells.

## Methods
**Mice**. Mice were bred in pathogen-free conditions, and care and use of animals were performed according to the European and national regulations, published in the Official Journal of the European Union L 276/33, September 22, 2010. All animal procedures were approved by the Landesamt für Gesundheit und Soziales (LaGeSo) Berlin with the number G101/18. Mice were housed in IVC cages type II at 21 °C ambient temperature, 50–60% humidity, and a 12-h dark/light cycle. Transgenic mouse lines used have been previously described: Lgr5-EGFP-IRES-CreERT2[16], β-catGOF[29], Gt(ROSA)26Sortm1Sor[31], Mll1flox[30]. Mutagenesis was induced in 4–6-weeks-old mice by intraperitoneal injections of tamoxifen (Sigma, 50 mg/kg, diluted 1:10 in sunflower oil) on three consecutive days. Mice were analyzed at the indicated time points after the last tamoxifen injection. Mice were given i.p. injections of BrdU (Millipore), final concentration 50 µg/g of body weight in PBS at 2 h before sacrifice. Kaplan–Meier plot for tumor incidence was calculated based on macroscopic observation of hyperplastic intestinal tissue. Both females and males were analyzed.

**Isolation of Lgr5-GFP[+] intestinal stem cells and RNA sequencing**. For isolation of Lgr5[+] stem cells, crypts were dissociated into single cells with TrypLE Express (ThermoFisher Scientific) for 30 min at 37 °C. Dissociated cells were passed through a 70 µm cell strainer and washed wit 5% FBS/PBS. Cells were stained with APC-conjugated anti-326 (EpCAM) antibody (1:100 dilution, eBioscience, Cat no. 17-5791-80), Alexa-Fluor 700 CD45 antibody (1:50 dilution, BD, Cat no. 560693) and PE-conjugated anti-CD24 (1:100 dilution, eBioscience, Cat. no. 12-0242-81) for 45 min on ice. Stained cells were collected by centrifugation and resuspended in SYTOX blue dead cell stain (1:20,000 dilution, ThermoFisher Scientific). FACS sorting was performed on a FACS AriaTM III cell sorter (BD). 300 single cells were sorted into a PCR tube containing 2 µl of nuclease-free H2O with 0.2% Triton-X

100 and 4 U murine RNase Inhibitor (NEB), and stored at −80 °C. RNA isolation and library preparation were performed based on the Smart-seq2 protocol[62]. In brief, RNA was denatured for 3 min at 72 °C in the presence of 2.4 mM dNTP (Invitrogen), 240 nM dT-primer, and 4U RNase Inhibitor (NEB), and reverse transcribed using the Superscript II Reverse Transcriptase (Invitrogen). Single-stranded cDNA was amplified with the Kapa HiFi HotStart Readymix (Roche) and purified using Sera-Mag SpeedBeads (GE Healthcare). cDNA quality and concentration were determined with the Fragment Analyzer (Agilent). Sequencing was performed on a NextSeq 500 (Illumina) with a sample sequencing depth of 30mio reads on average. RNA seq reads were aligned to the mm10 transcriptome with GSNAP (version 2018-07-04). Read counts for each gene were extracted using featureCounts (version 1.6.3). A table of read counts per gene was generated based on the overlap of uniquely mapped reads with the Ensemble Gene annotation (version 92). Read counts were subjected to statistical analysis using the DESeq2 R package (version 1.22.2). Euclidean distance analysis of the normalized counts was performed to compute the sample-to-sample correlation. Read counts from different biological groups were subjected to differential expression analysis with DESeq2, a maximum of 10% false discoveries (10% FDR) was accepted. Volcano plots were generated by a generic R $X-Y$ plotting using the $\log_2$FoldChange versus the $\log_2$ adjusted $p$-values. RNA seq differential expression data can be found in Supplementary Data 1.

**Organoid culture**. Intestinal organoids were obtained as previously described[63]: small intestines of mice were dissected and dissociated in 8 mM EDTA/PBS for 5 min at RT followed by 20 min incubation in ice-cold 2 mM EDTA/PBS at 4 °C. The epithelia were fractionated by shaking in ice-cold PBS. 250 crypts were seeded in 20 µl of growth factor-reduced Matrigel (BD Matrigel #356231) and cultured in basic crypt medium (60/40 Advanced DMEM/F12 supplemented with N2 and B27, GlutaMax, N-Acetylcysteine and Penicillin/Streptomycin (Invitrogen)) containing 50 ng/ml mEGF (Gibco), 100 ng/ml mNoggin (Peprotech) and 500 ng/ml hR-Spondin1 (Peprotech). Organoids were split every 4–6 days by mechanical disruption. Cre-mediated recombination was induced by the addition of 800 nM 4-hydroxy-tamoxifen for 2 days. β-cat[GOF] organoids were selected by R-Spondin1 withdrawal. All analyses were performed at 7–10 days after mutagenesis. Recombinant human BMP4 (Peprotech) was added for 24 h in Noggin-free crypt medium supplemented with EGF and R-spondin1. For serial re-plating, β-cat[GOF]; Mll1[−/−] organoids were dissociated into single cells with TrypLE Express (ThermoFisher Scientific) for 5 min at 37 °C, and 2000 cells were seeded in 20 µl of growth factor-reduced Matrigel and cultured in crypt medium supplemented with mEGF, mNoggin, and hR-spondin1.

**Cell culture**. HEK293TN cells and DLD1 and Ls174T human colon cancer cell lines were cultured in 1xDMEM supplemented with 10% fetal bovine serum (FBS) and 1%Penicillin/Streptomycin (PenStrep) at 37 °C and 5% CO2. Cell line identity was confirmed by Multiplex human Cell line Authentication (MCA, Multiplexion). For non-adherent sphere cultures, dishes were coated with 12 mg/ml polyhema/95% EtOH (Sigma) overnight at 55 °C. Single cells were cultured in CSC medium (DMEM/F12 (1:1) + GlutaMAX (Gibco), 1%PenStrep, supplemented with N2, 10 ng/ml bFGF (Gibco) and 10 ng/ml hEGF (Peprotech)) for 4–5 days, then dissociated with Trypsin for 5 min at 37 °C, and re-plated in CSC medium on polyhema-coated dishes. After 1 day of secondary sphere culture, spheres were harvested by centrifugation and processed for RNA/protein analysis. For serial re-plating assays, cells were seeded at a density of $0.5 \times 10^6$ cells on 10cm-dishes, trypsinized and counted every 2-3 days, and re-plated at a density of $0.5 \times 10^6$ cells. Gsk126 was administered for 96 h at 5 µM (McCabe et al. 2012)[54]. Tcf/Lef luciferase reporter assay (TOPflash) was performed as previously described[44], 3 µM CHIR99021 (Axon #1386) and 50 µM LF3[46] were added for 24 h.

**Generation of lentiviral particles and lentiviral transduction**. Doxycycline-inducible shRNA knockdown cell lines were generated by lentiviral transduction, using the doxycycline-inducible pInducer11 system[64]. shRNA targeting MLL1 was designed with the optimized miR30a backbone[65] (sequence shMLL1: TGCTGTT GACAGTGAGCGAAAGAAAGATTCTAAAAGTATATAGTGAAGCCACAGAT GTATATACTTTTAGAATCTTTCTTCTGCCTACTGCCTCGGA). For the production of lentiviral particles, HEK293TN cells were co-transfected with 10 µg pPAX.2, 2.5 µg pMD2.G and 10 µg pInducer11-shMLL1 by transfection with PEI (Sigma). Supernatants containing lentiviral particles were collected at 24 and 48 h after transfection. For lentiviral transduction, the lentiviral particle-containing supernatant was mixed 1:1 with a fresh growth medium containing 8 µg/ml polybrene. Ls174T and DLD1 cells were transduced at 300 g for 1 h at RT followed by incubating overnight at 37 °C. Transduced cells with high expression of shRNA were selected by fluorescence-activated cell sorting (FACS) for GFP and tRFP[64] and cultured as single-cell clones to establish shMLL1 cell lines. Knockdown was induced by the addition of doxycycline to the growth medium at a final concentration of 300 ng/ml.

**Xenografts**. $1 \times 10^6$ cells in 200 µl PBS were injected under the skin of nude mice, NMRI:nu/nu, female. In the case of shMLL1 cells, the inducible shMLL1 lines were pre-treated with 300 ng/ml doxycycline for 4 days, and mice were administered

doxycycline in drinking water (2 mg/ml doxycycline in 5 g/l sucrose) from day −1 throughout the experiment. Tumor size and mouse body weight were measured every other day.

**Histology, immunofluorescence staining, and analysis.** Analysis of human naïve colon cancer biopsies was approved by the ethic commission of the Friedrich-Alexander Universität Erlangen-Nürnberg (148_19 BC). Informed consent was obtained from all patients. Murine tissue was fixed in 4% formaldehyde/PBS, and histological analyses were performed on 5–7 μm paraffin sections. Organoids were fixed in 4% formaldehyde/PBS for 1 h and embedded in 1.5% agarose/PBS. For LacZ whole-mount staining, murine intestine was cut open longitudinally, fixed for 2 h in ice-cold fixative (1% formaldehyde, 0.2% glutaraldehyde, 0.02% NP-40 in PBS), washed twice in PBS and incubated in staining solution (2 mM MgCl$_2$, 5 mM K$_3$Fe(CN)$_6$, 5 mM K$_4$Fe(CN)$_6$, 0.01% NP-40, 0.01% sodium deoxycholate, 1 mg/ml X-gal (Roth) in PBS) overnight at RT protected from light. 10 μm paraffin sections of LacZ-stained tissue were counter-stained with nuclear fast red. For Mll1 immunostaining, antigen retrieval was performed by boiling in 10 mM sodium citrate pH 6.0 for 15 min, and sections were permeabilized in ice-cold methanol for 10 min prior to incubation in blocking solution (0.1% Tween20, 10% horse serum, 1% BSA in PBS) followed by incubation of the primary antibody overnight at 4 °C. Fluorochrome-conjugated or HRP-coupled secondary antibodies were incubated for 1 h at RT. Immunohistochemistry was developed with the DAB chromogenic substrate (DAKO).

**Light microscopy and data analysis.** Representative z-stacks were acquired with inverted laser scanning microscopes LSM710 and LSM700 using 405, 488, 561, and 633 nm lasers and a PlanApochromat ×40 NA 1.3 objective (Zeiss Jena, Germany) or a spinning disc confocal microscope CSU-W1 (Nikon/Andor) equipped with an iXON888 camera, using PlanApo ×20 NA 0.75 and Apo LWD ×40 NA 1.15 objectives. Images were processed using the software Zeiss ZEN (blue edition). Brightfield images were acquired with a Leica DIM6000 microscope equipped with a NPlan ×10 NA 0.25 and a motorized LMT200 V3 High precision Scanning Stage, using the Leica Application Suite X (LAS X). Images were analyzed with Fiji ImageJ. Mean fluorescence intensity in 3D reconstructed samples was quantified with Imaris 8 (Bitplane/Andor) software, using the surface module and selecting for nuclear fluorescence signal with a mask created in the DAPI channel. Mll1 staining intensity in the crypt cell populations was quantified with the measurement point tool in Imaris 8 and normalized to the staining intensity in the TA cell population.

**Chromatin immunoprecipitation.** Chromatin immunoprecipitation (ChIP) of histone modifications was performed from pInd11-shMLL1 Ls174T cell lines induced with 300 ng/ml doxycycline for 11d, following the instructions of the iDEAL ChIP-seq kit for histones (Diagenode). For ChIP of Mll1, TCF4, and β-catenin, pInd11-shMLL1 Ls174T and DLD1 cell lines were induced with 300 ng/ml doxycycline for 6d, and chromatin was prepared using the ChIP-IT Express kit (Active Motif). Cells were grown to 80% confluency, trypsinized for 3 min at 37 °C, fixed in 1% formaldehyde for 10 min, and quenched in glycine for 5 min at RT. Chromatin was sheared with a Branson Sonifier 450 (3 min shearing time, duty cycle 60, output control 6, sonified 10× for histone ChIPs, 4× for Mll1 ChIPs, 1 min pause between each sonication round). Shearing efficiency was checked on a 1% agarose gel. 10 μg of sheared chromatin was used for Mll1 ChIPs. ChIP-qPCR analysis was performed in a total volume of 20 μl SYBR green reaction mix (Roche Diagnostics) containing 0.25 μM of forward and reverse primers each in a CFX96-C1000T thermal cycler (BioRad): 2 min at 50 °C and 2 min at 95 °C followed by 42 cycles of 15 s at 95 °C and 1 min at 60 °C. Ct values of precipitated DNA were calculated relative to input DNA (% input). ChIP-qPCR primers were designed using H3K4 methylation profiles available in the UCSC genome browser (human reference genome GRCh37/hg19) and the Mll1 ChIP-seq UCSC genome browser data set from Active Motif (https://www.activemotif.com/catalog/details/61295/mll-hrx-antibody-pab). Primer sequences used for ChIP-qPCR are given in Supplementary Information (Supplementary Table 1).

**Western blotting.** Organoids were harvested and washed once in ice-cold 0.1% BSA-PBS, and cells were washed twice in PBS prior to lysis in ice-cold RIPA buffer (50 mM Tris pH8.0, 150 mM NaCl, 0.1% SDS, 1% NP-40, 0.5% sodium deoxycholate) containing protease inhibitors (cOmplete Mini EDTA-free, Roche). Total cell extracts were separated on polyacrylamide gels and transferred to a nitrocellulose membrane via semidry transfer for 1 h 15 min at 90 mA or wet transfer for 3 h at 85 V and 4 °C. Membranes were blocked with 5% BSA or 5% skim milk in 0.1% Tween20/TBS and probed with primary antibody diluted in blocking solution overnight at 4 °C. HRP-conjugated secondary antibodies were incubated for 1 h at RT. Immunoblots were developed with Western Lightning Plus ECL (Perkin Elmer) for 3 min and imaged with a Vilber Lourmat imaging system FUSION SL-3. Uncropped scans of Western blots are provided in the Source Data file.

**Antibodies.** The following antibodies were used in this study (dilutions given for immunostaining/Western Blot): anti-Mll1 (D6G8N, Cell Signaling #14197, 1:100/1:2000, 1:50 for ChIP), anti-hSet1 (Bethyl A300-289A, 1:2000), anti-H3K4me3

(Cell Signaling #9727, 1:2000/1:10,000, 1:50 for ChIP), anti-H3K27me3 (Millipore 07-449, 1:1000, 5 μg for ChIP), anti-H3K4me2 (Millipore #07-436, 1:5000), anti-H3K4me1 (Millipore #07-030, 1:5000), anti-H3 (Abcam ab1791, 1:10,000), anti-p21 (Santa Cruz sc-6246, 1:100/1:1000), anti-cleaved Caspase-3 (Cell Signaling #9661, 1:400/1:2000), anti-E-cadherin (BD 610181, 1:200), anti-Mmp7 (Santa Cruz, 1:100), anti-GATA4 (Santa Cruz Biotechnology sc-25310, RRID:AB_627667, 1:200), anti-GFP (Abcam ab6673, 1:500), anti-Ki67 (ThermoFisher MA5-14520, 1:300), anti-BrdU (Abcam ab6326, 1:100), anti-β-catenin (BD 610153, 1:300/1:1000, 3 μg for ChIP), anti-TCF4 (Cell Signaling #2569, 1:50 for ChIP), anti-Lyz (DAKO A0099, 1:500), anti-ITF (Santa Cruz sc-18272, 1:300), anti-alpha Tubulin (Sigma, 1:10,000), and anti-Vinculin antibody (Sigma V9131, 1:5000). For immunofluorescence and immunohistochemistry, cyanine-labeled secondary antibodies (Jackson ImmunoResearch) and HRP-conjugated polymer and DAB reagent (DAKO) were used. ChIP control antibodies were rabbit monoclonal IgG isotype control (Cell Signaling #3900) and mouse monoclonal IgG isotype control (Santa Cruz sc-2025). The specificity of histone modification antibodies was confirmed by dot blotting against synthetic peptides carrying the modifications of interest (Diagenode).

**RNA preparation for qRT-PCR analysis and RNA sequencing.** Total RNA was isolated by Trizol extraction (Invitrogen) or with the NucleoSpin RNA isolation kit (Macherey–Nagel). DNA contaminations were removed by DNase1 digestion (Invitrogen) in the presence of RNase inhibitor (RNase Out, Invitrogen), and RNA was purified via phenol/chloroform extraction. For qRT-PCR, 5 μg of total RNA were reverse transcribed with random hexamer primers (Invitrogen) and MMLV reverse transcriptase (Promega, 200 U/μl). qRT-PCR was performed in a total volume of 20 μl SYBR green reaction mix (Roche Diagnostics) containing 0.25 μM of forward and reverse primers each in a CFX96-C1000T thermal cycler (BioRad): 2 min at 50 °C and 2 min at 95 °C followed by 42 cycles of 15 s at 95 °C and 1 min at 60 °C. All reactions were performed as duplicates. Expression of target genes in treated versus control samples relative to the endogenous reference GAPDH was calculated using the ΔΔC$_t$ method. Primer sequences used for qRT-PCR are listed in Supplementary Information (Supplementary Table 2).

For RNA sequencing, total RNA from shMLL1 Ls174T cells induced with 300 ng/ml doxycycline for 3 days and from non-induced parental cells was isolated with the NucleoSpin RNA isolation kit (Macherey–Nagel) and sequenced on a NextSeq 500 (Illumina). RNA seq reads were quality-checked by FASTQC (v0.11.5) software. Sequencing reads were mapped to the human whole genome (hg38) using STAR aligner (v. 2.5.3a) and default parameters. Read counts for each gene (gencode v12) were extracted from the BAM file using featureCounts software (1.5.1). Read counts from different biological groups were subjected to differential expression analysis using the DESeq2 R statistical package. Comparisons between shMLL1 and parental control cells were corrected by individual technical replicates (paired comparison). In order to avoid background signal/noise, genes with <10 reads over all samples were excluded before adjusting the p-value for multiple testing. Genes with an adjusted p-value lower than 0.05 and absolute fold-change (log10) higher than 0.5 were considered differentially expressed. Gene expression profile comparison was performed by GeneOverlap R package by overlapping lists of differentially expressed genes (MGI symbol) from different sources (Clevers[41]; Soshnikova[42]; Battle[20]) with our own data. All conversions from Ensembl ID to MGI symbol and from human gene names to mouse gene names were performed by biomaRt. Volcano plots were generated by a generic R X–Y plotting using the log$_2$FoldChange versus the log$_2$ adjusted p-values. RNA seq differential expression data can be found in Supplementary Data 2.

**Human CRC transcriptomic data sets.** Transcriptome analyses in human CRC were carried out on 2207 samples that were available in two data repositories (Supplementary Table 3): (i) RNA seq (version 2) data from The Cancer Genome Atlas (TCGA) project (http://cancergenome.nih.gov/) were downloaded from the legacy version of the NCI Genomic Data Commons (GDC) repository[66]. Clinical and follow-up information was retrieved from the TCGA Pan-Cancer Clinical Data Resource (CDR) for outcome analytics[67]. A total of 543 TCGA CRC samples were available for analysis from this repository. (ii) Eight publicly available Affymetrix microarray data sets were downloaded from the NCBI GEO repository[68]. These data sets included gene expression and clinical information from a total of 1690 patients with CRC. In addition, clinical information for these data sets was downloaded from the Synapse repository[69] as provided by Guinney et al.[70] (syn2623706), and used to complete missing data in the retrieved, when possible.

Processing of TCGA RNA seq data: Four different TCGA data sets were processed separately, as colon and rectum samples had been sequenced independently using two different platforms (Genome Analyzer and HiSeq). For each of them, RSEM gene expression estimates were log2-transformed and quantile normalized. Only primary tumors from patients with no previous cancer diagnosis were kept for analyses. Patients with samples measured in the two platforms were excluded from the GA version. Samples TCGA-A6-2679-01A and TCGA-AA-A004-01A were also excluded, as their gene expression showed an anomalous distribution as compared to the rest of the samples in their data set, even after quantile normalization. A total of 543 TCGA CRC samples was available for analysis.

Processing of tumor microarray samples: Microarray samples were processed separately for each data set using packages affy[71] and affyPLM[72] from Bioconductor[73]. Raw CEL files were normalized using RMA background correction and summarization[74]. Standard quality controls were performed in order to identify abnormal samples and relevant sources of technical variability[75] regarding (a) spatial artifacts in the hybridization process (scan images and pseudo-images from probe-level models); (b) intensity dependences of differences between chips (MvA plots); (c) RNA quality (RNA digest plot); (d) global intensity levels (boxplot of perfect match log-intensity distributions before and after normalization and RLE plots); (e) anomalous intensity profile compared to the rest of samples (NUSE plots, Principal Component Analysis). Technical information concerning sample processing and hybridization was retrieved from the original CEL files: date of scanning were collected in order to define scan batches in each data set separately, and previously described technical metrics[76] were computed and recorded as additional features for each sample. Probesets were annotated using the information available on the Affymetrix-Thermofisher web page (https://www.thermofisher.com/es/es/home.html). No samples were excluded due to quality issues, leaving a total of 1690 microarray samples available for analysis.

Calibration and integration of tumor transcriptomic data: To make their expression values comparable, microarray data sets were corrected by potential sources of technical biases. For doing so, a linear model was fitted to each probeset and data set separately, in order to correct expression values by the center of origin of the sample, batch (scanning day), and probeset differences captured by PM.IQR, RMA.IQR and RNA.DEG metrics[76]. This correction was carried out using a mixed-effect model in which microsatellite instability (MSI) status was also included as a covariate. Scanning day was modeled as a random effect in these models. Expression intensities were summarized at the gene level (entrez) by the first principal component of the probesets mapping to the same gene. This component was centered and scaled to the weighted mean of the probesets' means and standard deviations, where the contributions to this first component were used as weights. The sign of this score was then corrected so that it was congruent to the sign of the probeset contributing the most to the first component. TCGA RNA seq expression data were processed in an analogous way; potential biases captured by sample's center and batch (plate) of origin were corrected using a mixed-effects model for each gene and data set separately. In such models, the center was modeled as a fixed effect while the plate identifier was included as a random effect. The R package lme was used in this procedure. Data sets were merged after quantile normalization and gene-wise standardization (median and median absolute deviation) to the GSE39582 data set according to their distribution of gender, age, site, MSI, and stage, when available. Such standardization was carried out by inverse probability weighting using logistic regression (LogReg) model. For doing so and for each data set D, a LogReg model was carried out on D and GSE39582 that included the aforementioned clinical parameters as predictors of the data set affiliation. Clinical parameters were included in the model as they were informative according to the Akaike Information Criterion (AIC) after running a forward step-wise algorithm for variables selection. Next, conditional probabilities of belonging to data sets D and GSE39582 (Pd and Pr, respectively) were retrieved from the model; then, weights were computed as the ratio of Pd/Pr; finally, weights were truncated to value "five", in order to avoid the overrepresentation of any sample in the standardization process. Expression values were truncated to the maximum and minimum values observed in this reference data set. Samples with unknown tumor stage information were excluded from the data. As a result, this CRC meta-cohort included a total of 2207 primary tumor samples for analyses.

Association of gene expression with clinical data in human CRC samples: Association with gene expression levels was assessed using Cox proportional hazards models (relapse and overall survival) and a linear model (stage). Statistical significance was assessed by means of a Log-likelihood Ratio Test (Cox models) or F-test (linear model), while Wald tests were used for pairwise comparisons. Sample groups of low, medium, and high expression levels were defined using the tertiles of the corresponding intensity distribution. Accordingly, hazard ratios (HR), adjusted group means, and their corresponding 95% confidence intervals were computed as measures of association. For visualization purposes, Kaplan–Meier survival curves were estimated for groups of samples showing low, medium, and high gene expression, while boxplots and stripcharts were built to display expression levels by tumor stage.

Only samples from patients diagnosed at stage I, II, or III were considered for analyses of time to relapse, for a total of 1095 samples. Overall survival analyses in pre-metastatic (stages I–II–III) and metastatic (stage IV) tumors were carried out on 981 and 165 samples, respectively. TCGA's rectum samples were excluded from these analyses accordingly to recommendations of the TCGA-Clinical Data Resource (CDR)[67]. The threshold for statistical significance was set at 5%. All analyses were carried out using R[77].

### Quantification and statistical analysis.
All data are presented as mean ± SD unless otherwise indicated. Statistical details of the experiments can be found in the figure legends. Graphs and statistics were generated with GraphPad Prism software. Tests for normal distribution were performed with D'Agostino–Pearson, and Shapiro–Wilk tests. Significance (p-values) was determined with the Mann–Whitney test (two-tailed) or two-tailed Student's t-test. No statistical method was applied to estimate sample size. No randomization or blinding protocol was used. Unless otherwise indicated, n reports the number of independent biological replicates per experiment. p-values ≤ 0.05 were considered statistically significant.

**Reporting summary**. Further information on research design is available in the Nature Research Reporting Summary linked to this article.

## Data availability
The RNA sequencing data from FACS-isolated Lgr5-GFP+ intestinal stem cells (Fig. 3b, d and Supplementary Fig. 3c, Supplementary Data 1) have been deposited in the Gene expression omnibus under accession number GSE148394. The RNA sequencing data from shMLL1 Ls174T colon cancer cells (Fig. 4c, d and Supplementary Fig. 4r, Supplementary Data 2) are available in the ArrayExpress database under accession number E-MTAB-8152. The human colon cancer data referenced during the study (Supplementary Table 3) are available in public repositories from the TCGA website (TCGA-COAD, TCGA-READ) and in the Gene Expression Omnibus (GEO) at NCBI (GSE14333, GSE33113, GSE39582, GSE38832, GSE44076, GSE13294, GSE18088 and GSE2109). Clinical information was obtained from Synapse ID syn2623706. All the other data supporting the findings of this study are available within the article and its Supplementary Information files and from the corresponding authors upon reasonable request. Source data are provided with this paper.

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

## Acknowledgements

We thank Russell Hodge (MDC) for helpful discussions and linguistic editing of the manuscript, Marcel Harrig (MDC), and Yvette Unger (MDC) for great reliability in maintaining the mouse colony, the Advanced Light Microscopy core facility of the MDC for assistance with imaging, H.-P. Rahn and the FACS core facility of the MDC for support with cell sorting, and Dimitra Alexopoulou and Andreas Dahl (DRESDEN-concept Genome Center, Center for Molecular and Cellular Bioengineering, Technische Universität Dresden) for the RNA sequencing. The results published here are in part based upon data generated by the TCGA Research Network: https://www.cancer.gov/tcga. This work was supported by MDC resources to W.B. and by funding from the Deutsche Forschungsgemeinschaft (STE903/7-3 to A.F.S.). J.G. was funded in part by the Berlin School of Integrative Oncology (BSIO) of the Charité Medical School Berlin.

## Author contributions

The project was conceived by J.H., J.G., and W.B. J.G., F.K., and J.H. performed the main experiments. N.G., A.K., A.W.-G., and D.B. conducted other experiments, R.O.V. performed bioinformatics analyses. J.G. and J.H. analyzed and interpreted the data. E.B. and A.B.-L. examined and interpreted human CRC patient gene expression and clinical data. A.F.S., S.S., M.V., and B.M. provided material. J.G., J.H., and W.B. wrote the manuscript. J.H. and W.B. supervised the project.

## Funding

## Competing interests

The authors declare no competing interests.
