## [Peer Review File · Nature Communications]

Reviewers' comments:

Reviewer #1 (Remarks to the Author):

In this work Grinat and colleagues have leveraged a mouse model and few more molecular models to investigate the role of the methyltransferase MLL1 in the context of cancer stem cells. I apologize preemptively if I have missed something here (as the manuscript is written in a rather convoluted way) but I would like to start by saying that the title itself is misleading as lot of the evidence reported here are applicable to the cell of origin of CRC (stem cell in the crypt) rather than specifically to cancer stem cells.

The authors start by characterizing WT mice villi for colocalization of LGR5 (the marker for normal stem cells in the crypt) and MLL1. The initial observation is intriguing and suggest that MLL1 expression is lost upon differentiation (both toward enterocytes or Paneth cell). One small note here, Figure 1D should be presented as not normalized (as it is not clear which TA cells they chose as a reference, it looks like n=4 but the number is much smaller than Paneth and Enterocytes). Furthermore, they should leverage the fact they sacrificed n=5 mice and quantify many more cells (especially stem cells).

In figure 2 the authors then used a Beta Catenin Gain of Function (GOF) that should stabilize B Catenin signaling and thus promote tumor formation. They then combined this model with MLL1 heterozygous and homozygous deletion. Here there is the first massive issue (that only emerge when discussing Figure 5). It is noticeable that MLL1 $-/-$ cells still have MLL1 (Figure 2B). I was puzzled about this result but then it becomes evident that MLL1 $-/-$ cells (and B catenin GOF) are lost from these animals rather early. Basically, if I understand this correctly, these mice revert to a WT phenotype as they lose all stem cells that have GPF and MLL1 $-/-$ (presumably because of the MLL1 loss). This is intriguing per se as it shows that MLL1 is required to maintain (some) stem cells. But the fact that the phenotype is not penetrant (quickly the intestine is taken over by WT cells) make these mice useless to say anything about tumor development. The KM analysis in Figure 5B is then tremendously misleading because (again, if I got this right) we are basically looking at B-catenin GOF vs. WT animals (and not MLL1 $-/-$). For the same reasons, all the experiment with this mouse model are extremely hard to interpret and I think have very little relevance to cancer biology. The authors in the discussion suggest that anti-MLL1 therapy might be an efficient way to target CRC. This is wrong and misleading because their data clearly show that cells that CSC that lose MLL1 are quickly replaced with WT ones (where wnt signaling can operate normally). It is clear to me that chemical targeting will be even less penetrant than a FLOX genetic system.

The second major issue comes up in Figure 3, when the authors attempt to identify a molecular mechanism. They quickly revert to an shMLL1 system and show that removing MLL1 selectively abrogates the expression of some Stem cell markers but not all of them (indicating that MLL1 has some sort of selectivity for promoters). They conclude that A) MLL1 is essential for the transcription of, say, LGR5 and B) there is some sort of B-Catenin MLL1 complex that recruit MLL1 at some genes but not others. There is no biochemistry done to support that MLL1 and TCF4/B-Catenin form complex on the chromatin. There is no rationale on why how MLL1 decides to methylate LGR5 but not SOX9 or AXIN2. But more importantly, I think the data have been misinterpreted. Their Figure 3G nicely show that removing H3K27me3 (that they suggest is downstream/consequence of loss of H3K4me3) is sufficient to re-activate LGR5. Unfortunately, the experiment was done in MLL1 shED cells (again, presumably, having very little MLL1). Hence, one might say that H3K4me3 is not needed to re-activate LGR5 and the key event is PRC2 mediated (fitting with the idea of differentiation and silencing of stem cell genes). Thus, I am tempted to conclude that the model the propose in 3H is rather misleading and not really supported by their data.

There are several other issues here and there but currently, I think it's not really useful to discuss them here. I apologize if there is something major, I have missed here, and I would be happy to see the responses to these criticism (that I hope might be constructive). A the current state, I fear

the manuscript is not ready for publication as there are too many critical issues with the experimental design, results and interpretation.

Reviewer #2 (Remarks to the Author):

This manuscript by Grinat et al., reports a functional role for the histone methyltransferase MLL1 in the regulation of both normal and neoplastic stem cells in the intestine. Mll1 is highly expressed in Lgr5 stem cells in the crypt and is necessary for the expansion and outgrowth of their transformed derivatives. Mechanistically, the authors implicate Mll1 in the control of a subset of critical intestinal stem cell genes through the maintenance of H3K4me3 at their transcriptional start sites. Overall, this work is of high quality, following a coherent logic and utilizing a variety of complementary approaches to probe the role of Mll1 in intestinal stem cells and cancer. This manuscript would be suitable for publication if the authors can address/clarify the following issues:

1. What was the impetus for studying Mll1 in the first place? Are other members of the Compass family expressed in the intestinal stem cell compartment?
2. The authors find that Mll1 is important for the transcription of various beta-catenin-regulated genes, including Lgr5. But Lgr5 is also important for the transduction of Wnt signaling? Therefore, what is the status of the Wnt signaling pathway in cells lacking Mll1? Is Mll1 itself controlled by Wnt signaling? The interplay between the Wnt pathway and Mll1, which could be bidirectional, is important to clarify here since it may impact how the other data is interpreted.
3. Knockdown of Mll1 has dramatic effects on the self-renewal of tumorspheres and on the ability of carcinoma cells to form tumors. Does Mll1 have an impact on cells when grown in 2D dimensional culture? Does Mll1 impact non-CSC populations? These experiments are needed in order to determine if loss of Mll1 is broadly cytotoxic/cytostatic or if its effects are truly specific to stem cells.
4. It would be nice to further substantiate the Mll1 staining in Figure 1 with transcript levels, especially for quantification. For example, how does the expression level compare between sorted Lgr5+ and Lgr5- populations?
5. The staining of clinical specimens is generally desirable but the relevance and interpretation of Fig 1E are questionable. Does Mll1 staining correlate with disease prognosis? Or is there a correlation with beta-catenin expression (which should be quantified) the important point here (e.g. see point 2 above)?
6. What exactly is the impact of Mll1 loss on the organoid growth in Figure 2? The authors state that "the expansion of the stem cells niches was strongly reduced..." with homozygous deletion of Mll1 but this is difficult to see from the figure. Is the number of organoids reduced? Their size? The presence of Lgr5+ cells?
7. The comparison to other intestinal stem cell signatures is important but Fig. 2C is confusing. Is the number presented in the squares or the Jaccard index more important? A better way to present this comparison would be helpful.
8. In Figure 6, the micrographs presented are all from Mll1-/- animals. A control group (either WT or Mll1+/-) should be included for comparison.
9. In the mouse intestine the authors find a correlation between Mll1 expression and global H3K4me3 whereas loss of Mll1 in Ls174T cells did not impact global levels H3K4 methylation. Can the authors comment on this discrepancy?
10. Does loss of Mll1 impact the Lgr5+ population in the absence of beta-catenin GOF? This would aid in understanding Fig 5, which presents very striking data. Does loss of Mll1 hamper the ability of activated beta-catenin to induce transformation? Or does it eliminate the target cell population that is subject to transformation?
11. What is the phenotype of the Mll1 knockout mice that were used here?
12. Line 227: "proceeded" should be "preceded."

Reviewer #3 (Remarks to the Author):

The manuscript focuses in the functional impact of Mll1 in ISC gene expression and homeostasis, and as consequence, in intestinal stemness and tumorigenesis. This is a very relevant issue, however, there are multiple concerns that diminish the significance of the presented data.

- 1) The only in vivo data that link Mll1 and ISCs is the IHC analysis of Mll1 in the mouse intestine. If Mll1 is relevant for normal ISC homeostasis, intestinal specific Mll1 deletion should impose massive intestinal defects, which should be easily tested using the appropriate mouse models. These experiments would reinforce the early impact of the work.
- 2) In addition, the possible correlation between nuclear beta-catenin and Mll1 levels in cancer (shown in 1e) needs to be quantitatively determined in a significant number of tumors.
- 3) Then authors move to the intestinal organoids model to demonstrate that Mll1 is required to support ISC gene expression and ISC expansion downstream of active β -cat. If Mll1 is required for Lgr5 and ISC gene expression, why Mll1 deletion prevents expansion of the Lgr5 population by β -catGOF, but does not affect Lgr5+ distribution in the (normal) crypt-like structures? Moreover, only one deleted organoid is shown in the figure; quantification of data, including the number of deleted organoids, and Lgr5+ cells per crypt unit in each condition is totally required, as well as a plausible explanation on how Mll1 deletion prevents β -catGOF expansion of the Lgr5+ population without affecting Lgr5+ in WT organoids (that similarly depends on beta-catenin activity).
- 4) It is also unclear whether cells expressing Lgr5 die following Mll1 deletion or they just stop proliferating as differences shown in S2b (60% proliferating WT Lgr5+ compared to 45% proliferating Mll1KO;Lgr5+ cells) cannot support the whole phenotype suggested in 2a and 2b.
- 5) In addition, it is at least surprising that organoids shown in 2a and 2b behave as monoclonal entities (individual organoids look as Mll1 all-negative or all-positive) if they don't derive from single ISCs (250 crypts seeded in 20 μ l of growth factor-reduced matrigel and then Split by mechanical disruption). If organoids are essentially polyclonal, one could anticipate that non-deleted cells will overgrow to replace Mll1-depleted stem cells. Can authors discuss this issue?
- 6) Then, authors generate inducible KD Ls174 cell clones and test them for gene transcription to conclude a general alteration of ISC signatures, which is subsequently linked to a reduction in organoid/tumoroid formation capacity. However, it is not mentioned whether adherent Ls174 cultures also show a proliferation or differentiation defect. This is important since this same cellular model is use for testing the effect of Mll1 deletion in Ls174 tumorigenic capacity. If Ls174 KD cells do not grow it is expected that tumor formation will be also impaired (no need of in vivo experiments). Moreover, the fact that tumors proliferate less does not support the idea that "Mll1 is required for "initiation" and growth of human Ls174T colon cancer xenografts".
- 7) In the section corresponding to figures 5 and 6, authors use an inducible Lgr5CreERT;b-catGOF;Mll1lox that to my opinion is the most convincing model (used in this manuscript) to demonstrate the requirement for Mll1 in beta-catenin induced tumorigenesis. Data included in this section is clear and demonstrate a relevant contribution of Mll1 to intestinal tumorigenesis downstream of beta-catenin. Nevertheless, in figure 6 (at the end of the results section) authors open the concept of mixed Paneth/goblet lineage imposed by Mll1 deletion and the suggestion that GATA6 and BMP are regulating ISC differentiation downstream of Mll1 (in supplementary data) but the results shown are very inconclusive and lacking quantification.

Thus, my general conclusion is that the work from Grinat al colleagues focuses in a relevant issue, which is the actual possibility that Mll1 epigenetically regulates ISC homeostasis and function. However, the data presented is very preliminary and fractional and do not to support the main assumptions of the manuscript. In my opinion, the work would be a good candidate for NatComm if authors center the experimental approach in the mouse model shown in figures 1 and 5, and elude doing fragmental experiments with random cellular models such as MEFs, DLD1, Ls174T cells and cell line-derived organoids.

Point-by-point response to the reviewers

Reviewer#1:

Question: *In this work Grinat and colleagues have leveraged a mouse model and few more molecular models to investigate the role of the methyltransferase MLL1 in the context of cancer stem cells. I apologize preemptively if I have missed something here (as the manuscript is written in a rather convoluted way) but I would like to start by saying that the title itself is misleading as lot of the evidence reported here are applicable to the cell of origin of CRC (stem cell in the crypt) rather than specifically to cancer stem cells.*

Answer: We would like to point out that we mainly address the role of Mll1 in the cell-of-origin in CRC in a genetic mouse model and translate our findings to human cancer stem cells. We changed the title to: "The epigenetic regulator Mll1 is required for Wnt-driven intestinal tumorigenesis and cancer stemness".

The authors start by characterizing WT mice villi for colocalization of LGR5 (the marker for normal stem cells in the crypt) and MLL1. The initial observation is intriguing and suggest that MLL1 expression is lost upon differentiation (both toward enterocytes or Paneth cell). One small note here, Figure 1D should be presented as not normalized (as it is not clear which TA cells they chose as a reference, it looks like n=4 but the number is much smaller than Paneth and Enterocytes). Furthermore, they should leverage the fact they sacrificed n=5 mice and quantify many more cells (especially stem cells).

To account for inter-animal variances in the staining efficiency and overall expression, we have normalized the staining intensity measured in the different cell types of n=5 individual mice to the particular expression level in TA cells. We have also adjusted our normalization and added the error bars to the different cell types. The data are normalized to the mean value of the TA cells located just above the stem cell niche (up to the relative +8 position). Further, the figure legend now specifies both that we have examined n=5 mice and that the quantification is based on several cells per crypt (Fig. 1f, page 30, lines 760-763).

In figure 2 the authors then used a Beta Catenin Gain of Function (GOF) that should stabilize B Catenin signaling and thus promote tumor formation. They then combined this model with MLL1 heterozygous and homozygous deletion. Here there is the first massive issue (that only emerge when discussing Figure 5). It is noticeable that MLL1 -/- cells still have MLL1 (Figure 2B). I was puzzled about this result but then it becomes evident that MLL1 -/- cells (and B catenin GOF) are lost from these animals rather early. Basically, if I understand this correctly, these mice revert to a WT phenotype as they lose all stem cells that have GPF and MLL1 -/- (presumably because of the MLL1 loss). This is intriguing per se as it shows that MLL1 is required to maintain (some) stem cells. But the fact that the phenotype is not penetrant (quickly the intestine is taken over by WT cells) make these mice useless to say anything about tumor development. The KM analysis in Figure 5B is then tremendously misleading because (again, if I got this right) we are basically looking at B-catenin GOF vs. WT animals (and not MLL1 -/-). For the same reasons, all the experiment with this mouse model are extremely hard to interpret and I think have very little relevance to cancer biology.

The β -cat^{GOF}; Mll1^{-/-} mice do not revert to a wild-type phenotype. We find wild-type crypts next to mutant crypts, which decline over time but do not entirely disappear. We demonstrated this by lineage tracing experiments and by quantifying the number of mutated (LacZ⁺) crypts, which increase in the β -cat^{GOF}; Mll1^{+/-} mice from 25% to 50%. In contrast, in β -cat^{GOF}; Mll1^{-/-} mice they gradually decrease from initial 25% to 10% (Supplementary Fig.

2e, f). We also introduced major changes in the way we present our data and changed the presentation of the gradual but not entire loss of the β -cat^{GOF}; Mll1^{-/-} crypts, see the persisting Mll1 knockout crypt next to non-recombined crypts in Fig. 2a, bottom right, with an inset for magnification. β -cat^{GOF}; Mll1^{-/-} crypts are found until 100 days after induction. These persisting Mll1-deficient crypts exhibit nuclear β -cat^{GOF}, but they do not form tumors (Supplementary Fig. 2h). This means that the β -cat^{GOF}; Mll1^{-/-} phenotype is persistent. The KM analysis confirms this (Fig. 2b, pages 6-7, lines 125-154).

The previous version of the manuscript showed recombined and non-recombined organoids. In new experiments, we now selected β -cat^{GOF} organoids by withdrawing R-spondin1. Control organoids did not grow in the absence of R-spondin1, while the β -cat^{GOF} mutation enabled R-spondin1-independent growth, regardless of the status of Mll1 (Supplementary Fig. 2k). Using immunofluorescence and RT-PCR (new Fig. 2e, Fig. 2g), we confirmed that the β -cat^{GOF}; Mll1^{-/-} organoids do not express Mll1. The loss of Mll1 prevents β -cat^{GOF}-induced stem cell expansion, as now also shown by Western blot quantification of Lgr5-GFP in organoids from the three mutant lines (Fig. 2f, pages 7-8, lines 155-166).

The authors in the discussion suggest that anti-MLL1 therapy might be an efficient way to target CRC. This is wrong and misleading because their data clearly show that cells that CSC that lose MLL1 are quickly replaced with WT ones (where wnt signaling can operate normally). It is clear to me that chemical targeting will be even less penetrant than a FLOX genetic system.

The reviewer is correct that the loss of Mll1 causes a gradual decline of β -cat^{GOF}-mutated intestinal stem cells/cancer stem cells over time, and that non-recombined (wild-type) stem cells repopulate the epithelia. However, our lineage tracing experiments clearly show that both genotypes produce mutant cells, and that Mll1 knockout crypts are not immediately lost (Supplementary Fig. 2e, f). Our data reveal that the loss of Mll1 prevents β -cat^{GOF}-induced stem cell expansion and tumor formation, demonstrating that Mll1 is a prerequisite for β -cat^{GOF}-mediated tumorigenesis (pages 6-7, lines 115-132). We have also isolated β -cat^{GOF}; Mll1^{-/-} stem cells using FACS. RNA-seq revealed that the loss of Mll1 reinforced differentiation of tumorigenic β -cat^{GOF}-mutated stem cells (new Fig. 3b), in other words, pushing them to less aggressive fates (page 9, lines 184-208). RNA-seq of MLL1-depleted human colon cancer cells confirmed these findings. The ablation of MLL1 in colon cancer sphere cultures, which enrich for stem-like cancer cells, downregulated the expression of stem cell genes and enforced differentiation (Fig. 4e, Supplementary Fig. 4n). These data show that interfering with MLL1 can eliminate the stemness of cancer cells by preventing the stemness-promoting effects of oncogenic Wnt signaling (pages 10-12, lines 235-280). Concerning cancer therapies, we believe that it is an advantage if cells with oncogene (β -cat^{GOF})-induced stemness are replaced by wild-type stem cells in which Wnt signaling can operate normally. It is known that wild-type stem cells, with functional Wnt signaling, are essential for intestinal homeostasis and do not expand into tumors¹⁻⁴.

The second major issue comes up in Figure 3, when the authors attempt to identify a molecular mechanism. They quickly revert to a shMLL1 system and show that removing MLL1 selectively abrogates the expression of some Stem cell markers but not all of them (indicating that MLL1 has some sort of selectivity for promoters). They conclude that A) MLL1 is essential for the transcription of, say, LGR5 and B) there is some sort of B-Catenin MLL1 complex that recruit MLL1 at some genes but not others. There is no biochemistry done to support that MLL1 and TCF4/B-Catenin form complex on the chromatin. There is no rationale on why how MLL1 decides to methylate LGR5 but not SOX9 or AXIN2. But more importantly,

I think the data have been mis-interpreted. Their Figure 3G nicely show that removing H3K27me3 (that they suggest is downstream/consequence of loss of H3K4me3) is sufficient to re-activate LGR5. Unfortunately, the experiment was done in MLL1 shED cells (again, presumably, having very little MLL1). Hence, one might say that H3K4me3 is not needed to re-activate LGR5 and the key event is PRC2 mediated (fitting with the idea of differentiation and silencing of stem cell genes). Thus, I am tempted to conclude that the model they propose in 3H is rather misleading and not really supported by their data.

Intestinal epithelia exhibit several Wnt-active cell types, such as Lgr5⁺ stem cells and Paneth cells, which selectively express different Wnt target genes^{5, 6}. Here we have identified Mll1 as a mediator of this differential gene expression. Mll1 controls the expression of the Wnt-dependent stem cell gene *Lgr5*⁶ and other stem cell genes to sustain stemness in an oncogene-driven system. The hypothesis that Tcf4/ β -catenin and Mll1 form a co-complex in intestinal cancer was based on our previous findings in salivary gland and head and neck cancer⁷.

Gsk126 treatment was indeed performed in shMLL1 knockdown cells. We confirmed that our shRNA system functions properly (Supplementary Fig. 4a-d), and that ultimately very little MLL1 is left. We now provide additional evidence for the effects of MLL1 on the stem cell gene regulation. Chromatin immunoprecipitation of H3K4me3 and H3K27me3 in control, shMLL1 and Gsk126-treated shMLL1 cells confirmed the removal of H3K27me3 at the *LGR5* promoter upon Gsk126 treatment, and revealed that levels of H3K4me3 remained unchanged (Fig. 6a, b). Despite the absence of MLL1 and H3K4me3, Gsk126 treatment partially restored *LGR5* expression in shMLL1 knockdown cells, showing that Mll1 and its methyltransferase activity are dispensable for transcriptional initiation, but critical for preventing polycomb-mediated repression of *LGR5* (page 13, lines 298-313). We observed the same for *OLFM4* (Fig. 6a, b). In contrast, the expression and H3K4me3/H3K27me3 levels of the classical Wnt target gene *AXIN2* and the Wnt-regulated stem cell gene *ASCL2* were not affected by MLL1 knockdown or Gsk126 treatment, suggesting that these genes are regulated independently of this Mll1-polycomb interrelation (Fig. 6c, d). RNA sequencing of FACsorted β -cat^{GOF}; Mll1^{-/-} stem cells revealed a differentiation of the β -cat^{GOF}-mutated stem cells upon ablation of Mll1 (Fig. 3b). The mechanism by which Mll1 promotes β -cat^{GOF}-driven intestinal tumorigenesis hence is the following: Mll1 sustains the expression of intestinal stem cell genes by preventing polycomb-mediated silencing. Mll1 thereby prevents stem cell differentiation, and upon a β -cat^{GOF} mutation promotes the expansion of the tumorigenic stem cells beyond the stem cell niche at the crypt base. Mll1 is thus essential for sustaining *Lgr5* expression and cancer stemness. We have now modified our model shown in Fig. 6e (page 13, lines 306-311).

There are several other issues here and there but currently, I think it's not really useful to discuss them here. I apologize if there is something major, I have missed here, and I would be happy to see the responses to these criticism (that I hope might be constructive). At the current state, I fear the manuscript is not ready for publication as there are too many critical issues with the experimental design, results and interpretation.

Overall the reviewer made critical comments about our manuscript but suggested a revision. We believe that our responses and the improvements of the manuscript will satisfy the reviewer's concerns regarding experimental design, results and data interpretation.

Reviewer#2:

This manuscript by Grinat et al., reports a functional role for the histone methyltransferase MLL1 in the regulation of both normal and neoplastic stem cells in the intestine. Mll1 is highly expressed in Lgr5 stem cells in the crypt and is necessary for the expansion and outgrowth of their transformed derivatives. Mechanistically, the authors implicate Mll1 in the control of a subset of critical intestinal stem cell genes through the maintenance of H3K4me3 at their transcriptional start sites. Overall, this work is of high quality, following a coherent logic and utilizing a variety of complementary approaches to probe the role of Mll1 in intestinal stem cells and cancer.

We thank the reviewer #2 for his/her positive opinion of our manuscript and experimental performance.

This manuscript would be suitable for publication if the authors can address/clarify the following issues:

1. *What was the impetus for studying Mll1 in the first place? Are other members of the Compass family expressed in the intestinal stem cell compartment?*

A RT-PCR analysis of crypt and villus fractions from wild-type intestines revealed that Mll1 and its family members are all expressed in the intestinal crypts (see Figure 1 below). Setd1a has previously been suggested to control colon cancer cell proliferation and metastasis⁸. Sphere cultures of human colon cancer cells, which enrich for self-renewing cancer stem cells, showed an increased expression of Mll1 but not its family member Setd1a (Supplementary Fig. 4h). Our lab had previously discovered a role of Mll1 in Wnt-driven salivary gland and head and neck cancer. The study by Zhu et al. (2019)⁷ further demonstrated a biochemical interaction of Mll1 with β -catenin. Given the strong dependence of colon cancer on Wnt signaling, we hypothesized that Mll1 might also play a role in Wnt-high colon cancer cells. We observed high Mll1 levels in the intestinal crypt compartment and in particular, in Lgr5⁺ stem cells, the cells-of-origin in colon cancer⁹ (Fig. 1e, f). Our observations of high Mll1 levels in human colon cancer biopsies corroborated our hypothesis (Fig. 1a) and were the impetus for studying Mll1 in a β -cat^{GOF}-driven mouse intestinal tumor model.

Figure 1: mRNA expression of Mll1 family members in crypt and villus fractions of the small intestinal epithelium, n=4 independent mice, unpaired t test.

2. *The authors find that Mll1 is important for the transcription of various beta-catenin-regulated genes, including Lgr5. But Lgr5 is also important for the transduction of Wnt signaling? Therefore, what is the status of the Wnt signaling pathway in cells lacking Mll1? Is*

Mll1 itself controlled by Wnt signaling? The interplay between the Wnt pathway and *Mll1*, which could be bidirectional, is important to clarify here since it may impact how the other data is interpreted.

Several lines of evidence demonstrated that the loss of *Mll1* does not affect the status of Wnt signaling. The ablation of *Mll1* did decrease the expression of the Wnt-regulated intestinal stem cell gene *Lgr5*, but there was no change in the expression of classical Wnt target genes such as *Axin2*¹⁰, *Sox9*¹¹ and the stem cell-specific transcription factor *Ascl2*¹². This differential regulation of Wnt targets was also observed in both human colon cancer cell lines Ls174T and DLD1 (Fig. 4e and Supplementary Fig. 4s, page 12, lines 273-278), and in β -cat^{GOF}; *Mll1*^{-/-} intestinal organoids (Fig. 2g, page 8, lines 157-167). β -cat^{GOF}; *Mll1*^{-/-} organoids could be maintained in the absence of the essential stem cell niche factor R-spondin1, demonstrating that the mutants had functional β -cat^{GOF} activity (Supplementary Fig. 2k). In β -cat^{GOF}; *Mll1*^{-/-} mice, mutant crypts lacking *Mll1* still exhibited nuclear β -catenin (Supplementary Fig. 2h). To confirm that the ablation of *Mll1* did not change Wnt activity, we also performed a Tcf4 reporter assay (Top/Fop) in MLL1-depleted human colon cancer cells: control and shMLL1 Ls174T cells showed equal levels of Tcf4 luciferase reporter activity (Fig. 4f, page 12, lines 278-280).

To investigate whether *Mll1* itself is controlled by Wnt signaling, we treated intestinal organoids from wild-type mice with Wnt3a (to stimulate Wnt signaling), then analysed *Mll1* expression. Wnt3a increased *Mll1* and *Lgr5* expression, whereas Bmp4 (to stimulate Bmp signaling and differentiation) decreased *Mll1* and *Lgr5* expression (Fig. 1g). This indicates that Wnt signaling promotes the expression of *Mll1*. A previous study of our lab had identified functional Tcf4 binding sites in the *Mll1* promoter⁷. Our data suggest that the expression of *Mll1* is high in Wnt-activated cells in a Bmp-low environment, which restricts *Mll1* expression to the intestinal crypts (page 6, lines 101-114). Accordingly, our data demonstrate that the interplay of the Wnt pathway and *Mll1* is not exactly bidirectional. *Mll1* controls selective Wnt target genes and mediates stemness induced by oncogenic Wnt, but does not globally alter Wnt signaling. The loss of *Mll1* is, however, sufficient to block tumor formation from Wnt-mutated stem cells (Fig. 2).

3. Knockdown of Mll1 has dramatic effects on the self-renewal of tumorspheres and on the ability of carcinoma cells to form tumors. Does Mll1 have an impact on cells when grown in 2D dimensional culture? Does Mll1 impact non-CSC populations? These experiments are needed in order to determine if loss of Mll1 is broadly cytotoxic/cytostatic or if its effects are truly specific to stem cells.

The knockdown of MLL1 did not affect the 2D growth or viability of the human colon cancer cell lines Ls174T and DLD1 over the first 3 passages after induction, i.e. up to 9 days on shMLL1 (now included in Supplementary Fig. 4f, g). This is in contrast to 3D tumor spheres, in which the ablation of MLL1 reduced secondary sphere formation by 80% in Ls174T and by 60% in DLD1 cells (Fig. 4a, Supplementary Fig. 4i). LacZ lineage tracing revealed the production of *Mll1*-null progeny (Supplementary Fig. 2e, f), and *Mll1* knockout crypts persisted up to 100 days after mutagenesis in the β -cat^{GOF}; *Mll1*^{-/-} mice (Fig. 2a-c, Supplementary Fig. 2i, j). MLL1 knockdown spheres and xenograft tumors were negative for the apoptosis marker cleaved Caspase-3 (Supplementary Fig. 4k, Supplementary Fig. 4m). Thus, our data show that the loss of *Mll1* does not have broad cytotoxic/cytostatic effects, but specifically affects oncogenic Wnt-driven stem cells (pages 10-11, lines 223-252).

4. *It would be nice to further substantiate the Mll1 staining in Figure 1 with transcript levels, especially for quantification. For example, how does the expression level compare between sorted Lgr5+ and Lgr5- populations?*

Our manuscript focuses on the cancer phenotype. So, our initial finding that Mll1 is highly expressed in the intestinal crypt (i.e. the Wnt-high compartment) and particularly in Lgr5⁺ stem cells is meant to solely associate Mll1 with stemness.

5. *The staining of clinical specimens is generally desirable but the relevance and interpretation of Fig 1E are questionable. Does Mll1 staining correlate with disease prognosis? Or is there a correlation with beta-catenin expression (which should be quantified) the important point here (e.g. see point 2 above)?*

The staining of biopsies from patients with colon cancer stages T0-T4 revealed no correlation between positive staining for Mll1 and tumor stage. Most tumors exhibited moderate to strong expression of Mll1 (Fig. 1a, Supplementary Fig. 1a). We do not have access to information on disease progression. But our immuno-histochemical analysis revealed a tight correlation of Mll1 and nuclear β -catenin. Tumors with high levels of nuclear β -catenin had high levels of Mll1 and vice versa (page 5, lines 76-83). We have now quantified this in all the specimens (Fig. 1b, Supplementary Fig. 1c) and demonstrated that *Mll1* expression is high in Wnt-activated cells (Fig. 1g). The correlation of high Mll1 and strong nuclear β -catenin is relevant to tumorigenesis, as we show in our mouse model of β -cat^{GOF}-driven intestinal tumors (Fig. 2).

6. *What exactly is the impact of Mll1 loss on the organoid growth in Figure 2? The authors state that “the expansion of the stem cells niches was strongly reduced...” with homozygous deletion of Mll1 but this is difficult to see from the figure. Is the number of organoids reduced? Their size? The presence of Lgr5+ cells?*

The loss of Mll1 had no effect on the number and size of organoids but prevented β -cat^{GOF}-driven increases in the number of Lgr5-GFP⁺ stem cells, i.e., it blocked the expansion of the stem cells and in consequence their niches. We re-phrased the text and included a Western blot of Lgr5-GFP expression in β -cat^{GOF}, β -cat^{GOF}; Mll1^{+/-} and β -cat^{GOF}; Mll1^{-/-} organoids compared to controls, which now provides a quantitative comparison of the expansion of the stem cell population (Fig. 2f, pages 7-8, lines 155-166).

7. *The comparison to other intestinal stem cell signatures is important but Fig. 2C is confusing. Is the number presented in the squares or the Jaccard index more important? A better way to present this comparison would be helpful.*

We like to keep this style of presenting the comparison, because it brings together two important points: the number of overlapping genes, on which our MLL1-regulated colon cancer stem cell signature is based, and the degree of similarity between the compared stem cell signatures (Jaccard index). A high number of overlapping genes does not necessarily indicate that two signatures are the most similar because of differences in the number of genes comprising each signature. ‘Soshnikova and shMLL1’ are less similar to each other than ‘Clevers and shMLL1’, despite the fact that the former pair shares a higher number of overlapping genes. The ‘Soshnikova’ signature contains many more genes than the ‘Clevers’ signature. The former Fig. 2c is now Fig. 4c in the new version of the manuscript.

8. *In Figure 6, the micrographs presented are all from Mll-/- animals. A control group (either WT or Mll+/-) should be included for comparison.*

Fig. 3c of the revised manuscript compares neighbouring crypts of β -cat^{GOF}; Mll1^{-/-} mice. In the same micrograph, we also show non-recombined Mll1-positive (left) and mutated Mll1-negative crypts (right), as seen through Mll1 staining. We believe that this does constitute a proper control, in the form of an adjacent crypt which is non-recombined. We have now also performed RNA sequencing on FACsorted Lgr5⁺ stem cells from β -cat^{GOF}; Mll1^{+/-} and β -cat^{GOF}; Mll1^{-/-} mice. The loss of Mll1 caused tumorigenic β -cat^{GOF} stem cells to differentiate and become secretory Paneth-goblet-like precursor cells, which we observe in the mutant crypts of β -cat^{GOF}; Mll1^{-/-} mice (Fig. 3b and c, page 9, lines 184-208). The former Fig. 6 is now Fig. 3 in the new version of the manuscript.

9. *In the mouse intestine the authors find a correlation between Mll1 expression and global H3K4me3 whereas loss of Mll1 in Ls174T cells did not impact global levels H3K4 methylation. Can the authors comment on this discrepancy?*

We observed a correlation between the distribution of Mll1 and H3K4me3 along the crypt-villus axis in the mouse intestine (Fig. 1d, Supplementary Fig. 1e, page 5, lines 89-99). We do not assume, however, that this necessarily indicates a functional association. Since the loss of Mll1 has no impact on the global levels of H3K4 methylation in Ls174T cells (Supplementary Fig. 5b), it is likely that other members of the Mll family that are expressed in the intestinal crypts contribute to the H3K4me3 pattern. These include the family member Setd1a, which functions as a H3K4 tri-methyltransferase^{13, 14}. The ablation of Setd1a has been suggested to cause a global reduction of H3K4me3 levels in colon cancer cells⁸.

10. *Does loss of Mll1 impact the Lgr5+ population in the absence of beta-catenin GOF? This would aid in understanding Fig 5, which presents very striking data. Does loss of Mll1 hamper the ability of activated beta-catenin to induce transformation? Or does it eliminate the target cell population that is subject to transformation?*

Supplementary Fig. 3 (previous version: Supplementary Fig. 5g) was intended to address this issue. We had used a LacZ reporter to trace the Mll1 knockout stem cells in Mll1^{-/-} mice without β -cat^{GOF} (Supplementary Fig. 3a). We have now also included immunohistochemistry staining for Mll1 to further substantiate the fate of Mll1 knockout crypts (Supplementary Fig. 3b). Our analyses showed that the loss of Mll1 in the non-tumorigenic Lgr5⁺ population has only a mild effect. The number of Mll1-knockout crypts decreased from initial 30-40% mutant crypts at day 10 down to 25-30% at day 50 after induction (Supplementary Fig. 3b). This indicates that non-recombined crypts slowly replaced Mll1^{-/-} crypts. A substantial proportion of Mll1^{-/-} crypts remained, however, demonstrating that the loss of Mll1 does not eliminate the target cell population that is subject to transformation (page 8, lines 173-179). In contrast, the loss of Mll1 in the β -cat^{GOF}-mutated Lgr5⁺ population had a strong effect on the β -cat^{GOF}-driven expansion of the mutant stem cells (Fig. 2c-f). The loss of Mll1 thus hampers the ability of β -cat^{GOF} to induce transformation, as seen by the fact that persisting β -cat^{GOF} stem cells that lack Mll1 are unable to expand and form tumors (Fig. 2a, b, pages 6-7, lines 125-154).

11. *What is the phenotype of the Mll1 knockout mice that were used here?*

We did not observe any effects of the Mll1 knockout on the life expectancy or health of the mice. In our Lgr5-EGFP-IRES-Cre^{ERT2}-based mouse model, the intestines remained intact and overall organ functions were not visibly impeded. The β -cat^{GOF} mutation did not produce tumors in the skin, stomach, lung, kidney or other organs with Lgr5⁺ stem cells.

12. *Line 227: "proceeded" should be "preceded."* We have corrected this.

Reviewer#3:

The manuscript focuses in the functional impact of Mll1 in ISC gene expression and homeostasis, and as consequence, in intestinal stemness and tumorigenesis. This is a very relevant issue, however, there are multiple concerns that diminish the significance of the presented data.

1) The only in vivo data that link Mll1 and ISCs is the IHC analysis of Mll1 in the mouse intestine. If Mll1 is relevant for normal ISC homeostasis, intestinal specific Mll1 deletion should impose massive intestinal defects, which should be easily tested using the appropriate mouse models. These experiments would reinforce the early impact of the work.

We like to point out that we did not carry out an in-depth study of the role of Mll1 in homeostasis of intestinal stem cells. Instead, our focus was directed toward the intestinal mouse tumor model to demonstrate that Mll1 plays an essential role in cancer stemness induced by oncogenic Wnt. We analysed the fate of Mll1-knockout intestinal stem cells in the absence of stabilized, oncogenic β -catenin: Mll1^{-/-} mice did not show any intestinal defects. Conditional mutagenesis using the *Lgr5-Cre^{ERT2}* causes a mosaic of recombined and non-recombined (wild-type) crypts. LacZ tracing revealed that mutant Mll1^{-/-} stem cells produce progeny (Supplementary Fig. 3a). Mll1^{-/-} crypts were progressively lost and functionally replaced by non-recombined crypts, but a substantial number of Mll1^{-/-} crypts persisted beyond 50 days after the induction of mutagenesis (Supplementary Fig. 3a, b). Our data indicate that, under normal homeostatic conditions, the loss of Mll1 causes a mild phenotype, with a slow reduction of mutant stem cells over time. On the other hand, under oncogenic conditions, we demonstrate that Mll1 becomes a crucial regulator, with a strong impact on tumor-initiating stem cells (Fig. 2c, Supplementary Fig. 2e, f, pages 6-7, lines 125-154 and page 8, lines 173-183).

2) In addition, the possible correlation between nuclear beta-catenin and Mll1 levels in cancer (shown in 1e) needs to be quantitatively determined in a significant number of tumors.

We have now quantified the levels of nuclear beta-catenin and Mll1 in n=8 human tumors and 5 different tumor stages. This shows that tumors with high nuclear β -catenin are also high in Mll1 across all tumor stages (new Fig. 1b, page 5, lines 76-83).

3) Then authors move to the intestinal organoids model to demonstrate that Mll1 is required to support ISC gene expression and ISC expansion downstream of active β -cat. If Mll1 is required for Lgr5 and ISC gene expression, why Mll1 deletion prevents expansion of the Lgr5 population by β -catGOF, but does not affect Lgr5+ distribution in the (normal) crypt-like structures? Moreover, only one deleted organoid is shown in the figure; quantification of data, including the number of deleted organoids, and Lgr5+ cells per crypt unit in each condition is totally required, as well as a plausible explanation on how Mll1 deletion prevents β -catGOF expansion of the Lgr5+ population without affecting Lgr5+ in WT organoids (that similarly depends on beta-catenin activity).

The reviewer raises an important point. The absence of Mll1 prevents the expansion of Lgr5-GFP⁺ stem cells by β -cat^{GOF}, but does not affect Lgr5-GFP⁺ stem cells in the crypt-based stem cell niches (Fig. 2c, d). Our data thus suggest that stem cells can maintain their stemness and the expression of stem cell genes without Mll1, as long as they reside in the stem cell niche at the base of the crypts. Treatment of MLL1-depleted human colon cancer cells with the polycomb/EZH2 inhibitor Gsk126 partially restored the expression of the Mll1-controlled stem cell genes *LGR5* and *OLMF4* even though MLL1 and H3K4me3 were absent (Fig. 6a, b). This means that Mll1 and its methyltransferase activity are dispensable for the transcriptional initiation, but crucial for sustaining stem cell gene expression by preventing polycomb-mediated silencing (page 13, lines 298-313). The β -cat^{GOF}-driven expansion of stem cells depend on their intrinsic ability to sustain stemness and the expression of stem cell genes. Our work identifies Mll1 as a critical mediator and a prerequisite of this β -cat^{GOF}-driven stem cell expansion and tumorigenesis. Mll1 is required to sustain β -cat^{GOF}-driven stem cell gene expression and cancer stemness independent of the stem cell niche in crypts. Wnt-mutated stem cells that lose Mll1 cannot expand niche-independently to form tumors. The loss of Mll1 did not affect global activity of oncogenic Wnt signaling, as β -cat^{GOF}; Mll1^{-/-} organoids could be maintained in the absence of the essential Wnt-stimulating niche factor R-spondin1 (Supplementary Fig. 2k). Serial re-plating of single cell-dissociated β -cat^{GOF}; Mll1^{-/-} organoids now provides evidence that the loss of Mll1 progressively exhausted the self-renewal capacity of β -cat^{GOF} stem cells (Supplementary Fig. 2l, m, page 8, lines 157-172).

To address the reviewer's concerns about the organoid experiments, we now select for β -cat^{GOF}-mutated organoids by withdrawing R-spondin1. Control (non-recombined) organoids did not grow in the absence of R-spondin1. However, the β -cat^{GOF} mutation permitted R-spondin1-independent growth (Supplementary Fig. 2k). Through immunofluorescence staining and RT-PCR, we confirmed that the β -cat^{GOF}; Mll1^{-/-} organoids did not express Mll1 (Fig. 2e, g). We have included a quantification of stem cells by Western blotting for Lgr5-GFP in β -cat^{GOF}, β -cat^{GOF}; Mll1^{+/-} and β -cat^{GOF}; Mll1^{-/-} organoids compared to control organoids to quantify the stem cell expansion phenotype (Fig. 2f).

4) *It is also unclear whether cells expressing Lgr5 die following Mll1 deletion or they just stop proliferating as differences shown in S2b (60% proliferating WT Lgr5+ compared to 45% proliferating Mll1KO;lgr5+ cells) cannot support the whole phenotype suggested in 2a and 2b.*

The slightly reduced proliferation cannot explain why the loss of Mll1 so strongly affects the expansion of β -cat^{GOF}-mutated stem cells. Mll1 deletion does not cause the death of Lgr5⁺ stem cells. Mll1 knockout stem cells potently produced recombined progeny, as revealed by LacZ lineage tracing, and there was no evidence of apoptosis in Mll1 knockout cells (Supplementary Fig. 2e, f, j). To gain deeper insights into the role of Mll1 in the β -cat^{GOF} stem cells, we have now performed RNA sequencing of FACsorted Lgr5⁺ stem cells from β -cat^{GOF}; Mll1^{+/-} and β -cat^{GOF}; Mll1^{-/-} mice. This revealed that β -cat^{GOF}; Mll1^{-/-} stem cells are forced into differentiation (Fig. 3b, pages 9-10, lines 184-222). Hence, Mll1 ablation affects β -cat^{GOF}-driven stem cell expansion in two ways: i) ablation of Mll1 enables unrestricted polycomb-mediated repression of key stem cell genes and prevents β -cat^{GOF}-induced stemness, and ii) Mll1 ablation reduces proliferation and enforces differentiation of the mutant stem cells, pushing them to less aggressive fates (pages 15-16, lines 340-376).

5) *In addition, it is at least surprising that organoids shown in 2a and 2b behave as monoclonal entities (individual organoids look as Mll1 all-negative or all-positive) if they don't*

derive from single ISCs (250 crypts seeded in 20 μ l of growth factor-reduced matrigel and then Split by mechanical disruption). If organoids are essentially polyclonal, one could anticipate that non-deleted cells will overgrow to replace Mll1-depleted stem cells. Can authors discuss this issue?

The reviewer is right that the mutant organoids appear to behave as monoclonal entities, although they are split by mechanical disruption and do not derive from single stem cells, thus producing a polyclonal organoid pool. In the organoids as well as *in vivo*, however, either completely recombined (all-positive) or non-recombined (all-negative) crypts were observed (Fig. 2a, e). Apparently, the β -cat^{GOF} activation gives mutant stem cells a growth advantage that allows them to take over the entire crypt. This is independent of the Mll1 status, as the loss of Mll1 does not impair the global activity of oncogenic Wnt signaling (Supplementary Fig. 2k, Fig. 4f). We have now selected for mutant organoids by withdrawing the essential growth factor R-spondin1 (new images in Fig. 2e, page 8, lines 157-159). Initially, β -cat^{GOF}; Mll1^{-/-} organoids grow in the absence of R-spondin1, demonstrating functional β -cat^{GOF} activity in the β -cat^{GOF}; Mll1^{-/-} cells. β -cat^{GOF}; Mll1^{-/-} mutant stem cells give rise to Mll1-negative progeny, rendering organoids and *in vivo* crypts 'Mll1 all-negative'. However, these Mll1-deficient cells fail to maintain their β -cat^{GOF}-induced stemness, cannot expand into tumors and lose their organoid forming capacity (Supplementary Fig. 2l, m, pages 7-8, lines 133-183).

*6) Then, authors generate inducible KD Ls174 cell clones and test them for gene transcription to conclude a general alteration of ISC signatures, which is subsequently linked to a reduction in organoid/tumoroid formation capacity. However, it is not mentioned whether adherent Ls174 cultures also show a proliferation or differentiation defect. This is important since this same cellular model is use for testing the effect of Mll1 deletion in Ls174 tumorigenic capacity. If Ls174 KD cells do not grow it is expected that tumor formation will be also impaired (no need of *in vivo* experiments). Moreover, the fact that tumors proliferate less does not support the idea that "Mll1 is required for "initiation" and growth of human Ls174T colon cancer xenografts".*

We have now included data on the adherent growth of MLL1 knockdown cells. Knockdown of MLL1 did not affect the 2D growth and the viability of the human colon cancer cell lines Ls174T and DLD1 over the first three passages after induction, i.e. up to 9 days (Supplementary Fig. 4f, g). In contrast, the ablation of MLL1 had an immediate effect on the growth and re-plating efficiency of 3D tumor spheres. Secondary sphere formation of MLL1 knockdown cells was reduced by 80% in Ls174T and by 60% in DLD1 cells (Fig. 4a, Supplementary Fig. 4i, pages 10-11, lines 223-242). The Ls174T cells for the xenograft experiments were pre-treated with shMLL1 in 2D culture for 3 days only. This is sufficient to knock down MLL1 at the RNA and protein levels, but does not affect the proliferation of the shMLL1 Ls174T cells upon cell inoculation (Supplementary Fig. 4a-d, f). We believe that the tumor formation from cells that have established the knockdown of Mll1 at the time of injection does reflect tumor initiation.

7) In the section corresponding to figures 5 and 6, authors use an inducible Lgr5CreERT;b-catGOF;Mll1lox that to my opinion is the most convincing model (used in this manuscript) to demonstrate the requirement for Mll1 in beta-catenin induced tumorigenesis. Data included in this section is clear and demonstrate a relevant contribution of Mll1 to intestinal tumorigenesis downstream of beta-catenin. Nevertheless, in figure 6 (at the end of the results section) authors open the concept of mixed Paneth/goblet lineage imposed by Mll1

deletion and the suggestion that GATA6 and BMP are regulating ISC differentiation downstream of Mll1 (in supplementary data) but the results shown are very inconclusive and lacking quantification.

We thank the reviewer for his/her advice to place a focus on the Lgr5-Cre; β -cat^{GOF}-driven intestinal mouse tumor model and the respective organoid cultures. We have now rearranged the structure of our manuscript so that these data are now found in Fig. 2 and 3. Additional data are presented on the mixed Paneth/goblet cells observed in the β -cat^{GOF}; Mll1^{-/-} crypts, associating these cells with the differentiation of β -cat^{GOF}-mutated Lgr5⁺ stem cells following ablation of Mll1. RNA sequencing of FACsorted Lgr5-GFP⁺ intestinal stem cells from β -cat^{GOF}; Mll1^{+/-} and β -cat^{GOF}; Mll1^{-/-} mice revealed that the β -cat^{GOF}; Mll1^{-/-} stem cells differentiated toward the secretory lineage (Fig. 3b, page 9, lines 184-208). Mll1-depleted stem cells exhibited an increase in the expression of Paneth/goblet cell genes. The RNA sequencing also showed a strong decrease in the expression of *Gata4*, the small intestinal homologue of *Gata6*, upon ablation of Mll1, which we confirmed by immunofluorescence (Fig. 3d, e, pages 9-10, lines 209-219). These data exposed *Gata4* as a Mll1-dependent gene in Lgr5⁺ intestinal stem cells.

Thus, my general conclusion is that the work from Grinat al colleagues focuses in a relevant issue, which is the actual possibility that Mll1 epigenetically regulates ISC homeostasis and function. However, the data presented is very preliminary and fractional and do not to support the main assumptions of the manuscript. In my opinion, the work would be a good candidate for NatComm if authors center the experimental approach in the mouse model shown in figures 1 and 5, and elude doing fragmental experiments with random cellular models such as MEFs, DLD1, Ls174T cells and cell line-derived organoids.

We appreciate the positive opinion of reviewer #3 on our work. We have now addressed all the concerns and restructured the manuscript to stress our focus on the mouse model. The additional data buttress the original findings from our study and strongly support our main assumptions and conclusions.

References for the reviewers:

1. van Es, J.H. *et al.* A critical role for the Wnt effector Tcf4 in adult intestinal homeostatic self-renewal. *Mol Cell Biol* **32**, 1918-1927 (2012).
2. Korinek, V. *et al.* Depletion of epithelial stem-cell compartments in the small intestine of mice lacking Tcf-4. *Nat Genet* **19**, 379-383 (1998).
3. Fevr, T., Robine, S., Louvard, D. & Huelsken, J. Wnt/beta-catenin is essential for intestinal homeostasis and maintenance of intestinal stem cells. *Mol Cell Biol* **27**, 7551-7559 (2007).
4. Pinto, D., Gregorieff, A., Begthel, H. & Clevers, H. Canonical Wnt signals are essential for homeostasis of the intestinal epithelium. *Genes Dev* **17**, 1709-1713 (2003).
5. Battle, E. *et al.* Beta-catenin and TCF mediate cell positioning in the intestinal epithelium by controlling the expression of EphB/ephrinB. *Cell* **111**, 251-263 (2002).
6. Barker, N. *et al.* Identification of stem cells in small intestine and colon by marker gene Lgr5. *Nature* **449**, 1003-1007 (2007).
7. Zhu, Q. *et al.* The Wnt-Driven Mll1 Epigenome Regulates Salivary Gland and Head and Neck Cancer. *Cell Rep* **26**, 415-428 e415 (2019).
8. Salz, T. *et al.* hSETD1A regulates Wnt target genes and controls tumor growth of colorectal cancer cells. *Cancer Res* **74**, 775-786 (2014).

9. Barker, N. *et al.* Crypt stem cells as the cells-of-origin of intestinal cancer. *Nature* **457**, 608-611 (2009).
10. Lustig, B. *et al.* Negative feedback loop of Wnt signaling through upregulation of conductin/axin2 in colorectal and liver tumors. *Mol Cell Biol* **22**, 1184-1193 (2002).
11. Blache, P. *et al.* SOX9 is an intestine crypt transcription factor, is regulated by the Wnt pathway, and represses the CDX2 and MUC2 genes. *J Cell Biol* **166**, 37-47 (2004).
12. van der Flier, L.G. *et al.* Transcription factor achaete scute-like 2 controls intestinal stem cell fate. *Cell* **136**, 903-912 (2009).
13. Ardehali, M.B. *et al.* Drosophila Set1 is the major histone H3 lysine 4 trimethyltransferase with role in transcription. *EMBO J* **30**, 2817-2828 (2011).
14. Wu, M. *et al.* Molecular regulation of H3K4 trimethylation by Wdr82, a component of human Set1/COMPASS. *Mol Cell Biol* **28**, 7337-7344 (2008).

REVIEWER COMMENTS

Reviewer #1 (Remarks to the Author):

The authors have attempted to provide an explanation to all the points raised in my first review. I believe the manuscript has been improved significantly but I also believe that reviewer 3 has raised some important concerns and they should be taken into account in the final decision.

Reviewer #3 (Remarks to the Author):

To my view, author made a clear effort to improve the manuscript and to answer most reviewers' concerns. However, I still have some questions that maybe authors can clarify or complete.

1- In new Figure 1b, there is no information about the number of tumors in each groups of beta-cat staining (week, moderate, high) and whether beta-cat and mll1 levels are associated to a particular tumor stage (which could then explain beta-cat/mll1 correlation). In addition, although differences are clear, there is no statistical analysis of the results. Even more relevant to reinforce the importance of this work would be including an extensive analysis of Mll1 distribution in human CRC (specifically at stages that novel biomarkers are clearly required such as stages II and III) to determine whether Mll1 has prognostic value by itself (which should be expected if Mll1 is required for tumor stemness) and could represent a valuable therapeutic target.

2- In response to my question on the selective requirement of Mll1 for transformed stem cells but not in normal stem cells authors argue "Our data thus suggest that stem cells can maintain their stemness and the expression of stem cell genes without Mll1, as long as they reside in the stem cell niche at the base of the crypts." However, I do not really know what represents this stem cell niche and whether this applies to the organoids system where the only possible stem cells niche is the Paneth component that is probably present also in the beta-cat GOF organoids but also in many human CRC tumor. LYZ1 staining (plus Mll1 and/or Lgr5) in the organoids and in human tumors would support a model in which only Lgr5 cells adjacent to Paneth Cells can grow in the absence of Mll1 (if I understood correctly).

Response to the referee

For clarity, all reviewer comments are shown in *italics*, with our responses below. **New parts of the manuscript that were included in the response are highlighted in yellow here as well as in the revised version of the manuscript.**

Point-by-point answers NCOMMS19-24549A-Z

Reviewer #3

To my view, author made a clear effort to improve the manuscript and to answer most reviewers' concerns. However, I still have some questions that maybe authors can clarify or complete. 1- In new Figure 1b, there is no information about the number of tumors in each groups of beta-cat staining (weak, moderate, high) and whether beta-cat and mll1 levels are associated to a particular tumor stage (which could then explain beta-cat/mll1 correlation). In addition, although differences are clear, there is no statistical analysis of the results. Even more relevant to reinforce the importance of this work would be including an extensive analysis of Mll1 distribution in human CRC (specifically at stages that novel biomarkers are clearly required such as stages II and III) to determine whether Mll1 has prognostic value by itself (which should be expected if Mll1 is required for tumor stemness) and could represent a valuable therapeutic target.

We have now included the requested information in Fig. 1b and its legend: we added the number of tumors in each group and a statistical analysis for significance of β -catenin-MLL1 correlation by chi-square test (page 32, line 782). In Supplementary Fig. 1a, we show that the level of MLL1 expression is not associated to any particular tumor stage. We have now included statistical analysis of the observed MLL1 expression in T0-T4 by linear regression to confirm this observation (Supplementary Fig. 1a, page 37, line 892). We have further added a quantification of the levels of nuclear β -catenin across all tumor stages, which shows high levels of nuclear β -catenin in a major portion of the tumors at all stages (new Supplementary Fig. 1e, page 37, lines 897-899). Tumors with high levels of nuclear β -catenin showed high levels of MLL1 at all tumor stages, there is no association to a particular tumor stage. In Fig. 1e, we show that Mll1 is highly expressed in Wnt-activated intestinal epithelial cells, suggesting that the correlation derives from a Wnt-dependent regulation of MLL1, as had also been suggested in Zhu et al. (2019)¹.

As suggested by the editor and reviewer #3, we established a collaboration with Eduard Batlle and his bioinformatician from IRB Barcelona, who have access to gene expression data sets of CRC patient cohorts with clinical data as demonstrated in Merlos-Suárez et al. (2011)². They analysed several transcriptomic CRC patient cohorts. **The majority of tumors exhibited moderate to strong MLL1 expression, which was not associated with any particular tumor stage. We used a large transcriptomic colon cancer patient dataset (n=1095) to evaluate the association of MLL1 with disease progression. These analyses show no significant changes in MLL1 expression across tumor stages (Supplementary Fig. 1b). Patients of stages I-III and stage IV were stratified in three groups based on the level of MLL1 expression (MLL1 low, medium and high, Supplementary Fig. 1c). High MLL1 levels were associated with decreased overall survival and increased risk of disease relapse (Supplementary Fig. 1c, d). Altogether, high MLL1 levels in colon cancer are associated with poor patient survival and correlate to high Wnt activity (see page 5, lines 86-98).**

2- In response to my question on the selective requirement of *Mll1* for transformed stem cells but not in normal stem cells authors argue “Our data thus suggest that stem cells can maintain their stemness and the expression of stem cell genes without *Mll1*, as long as they reside in the stem cell niche at the base of the crypts.” However, I do not really know what represents this stem cell niche and whether this applies to the organoids system where the only possible stem cells niche is the Paneth component that is probably present also in the beta-cat GOF organoids but also in many human CRC tumor. LYZ1 staining (plus *Mll1* and/or *Lgr5*) in the organoids and in human tumors would support a model in which only *Lgr5* cells adjacent to Paneth Cells can grow in the absence of *Mll1* (if I understood correctly).

The stem cell niche we refer to is orchestrated by mesenchymal stroma-derived Wnt ligands and Bmp antagonists, and the Paneth cells as direct epithelial neighbours of the stem cells. In small intestinal organoids, which represent pure epithelial cultures, stroma-derived Bmp antagonists such as Noggin must be added exogenously to promote organoid formation and the Paneth cells are the major determinants for Wnt gradient-dependent formation of the stem cell niche^{3, 4}. In contrast, *in vivo* Paneth cells are dispensable for stem cell maintenance^{5, 6}. Normal intestinal stem cells express stem cell genes as long as they reside within the Wnt-high stem cell niche, and silence stem cell gene expression when the cells exit the Wnt-high niche to differentiate. A β -cat^{GOF} mutation imposes high Wnt activity and a stem-like gene expression programme^{7, 8} independent of the niche factors. β -cat^{GOF} impairs the downregulation of stem cell genes when the transformed cells exit the niche. Thus, the β -cat^{GOF}-driven expansion of intestinal stem cells relies on the intrinsic ability of the transformed stem cells to sustain their oncogenic Wnt-driven stemness and the expression of stem cell genes outside of the crypt stem cell niche. We demonstrate that the histone methyltransferase *Mll1* is required to sustain β -cat^{GOF}-driven stem cell gene expression and cancer stemness. Our data reveal that *Mll1* does not initiate the expression of *Lgr5*, *Olfm4* and other stem cell genes but maintains their activity by preventing polycomb-mediated silencing and thus sustains cancer stemness (Fig. 6 in the revised manuscript, pages 13-14, lines 318-333). In the absence of *Mll1*, β -cat^{GOF} stem cells lose their niche-independent stemness and differentiate when they leave the Wnt-high niche (similar to normal stem cells), as the β -cat^{GOF}-induced expression of stem cell genes cannot be maintained. Consequently, the loss of *Mll1* shows a strong effect on the expansion of transformed stem cells but plays a minor role in maintenance of normal stem cells that are committed to lose their stemness upon exit from the stem cell niche.

The reviewer requested to address the role of Paneth cells in supporting stemness and growth of *Mll1*-deficient β -cat^{GOF} cancer stem cells in intestinal tumors. The Wnt ligand-producing Paneth cells are themselves controlled by Wnt signalling: high Wnt activity induces the maturation of Paneth cells in intestinal crypts and adenomas^{9, 10}. Hence, Paneth cells are present in Wnt-high human colon cancers. As proposed by the reviewer, we stained β -cat^{GOF} tumor sections for *Lgr5*-GFP (stem cells) and *Lyz* (Paneth cells). These stainings revealed that established β -cat^{GOF} tumors do contain Paneth cells but do not exhibit a strict alternation of Paneth cells and mutant *Lgr5*⁺ stem cells (Fig. R1 sent to reviewer: the expanding mutant stem cells are not all in direct neighbourhood to Paneth cells). We observed both *Lgr5*-GFP⁺ stem cell-rich regions interspersed with *Lyz*⁺ Paneth cells (inset a) and without *Lyz*⁺ Paneth cells (inset b), suggesting that mutant β -cat^{GOF} stem cells can maintain their stemness independent of a direct cell-to-cell contact to a Paneth cell.

Figure R1 sent to reviewers: Lgr5-GFP and Lyz staining on $\beta\text{-cat}^{\text{GOF}}$ intestinal tumor.

Immunofluorescence for Lgr5-GFP (green) and Lyz (red) on $\beta\text{-cat}^{\text{GOF}}$ intestinal tumor section. Tumors exhibit expansions of mutant Lgr5-GFP⁺ $\beta\text{-cat}^{\text{GOF}}$ stem cells. Insets a and b: magnifications of regions with and without Paneth cells (Lyz⁺, red) adjacent to mutant stem cells.

We would like to point out that we cannot test a model in which only Lgr5⁺ stem cells adjacent to Paneth cells can grow in the absence of Mli1, as the loss of Mli1 prevents the formation of tumors/tumorous organoids. $\beta\text{-cat}^{\text{GOF}}$; Mli1^{-/-} mice do not develop any tumors in which the role of the Paneth cell niche for the maintenance of Mli1-deficient cancer stem cells could be tested. The $\beta\text{-cat}^{\text{GOF}}$; Mli1^{fllox} organoids and the *in vivo* tumor model provide information on the role of Mli1 in the initiation of oncogenic Wnt-driven intestinal cancer.

In the organoids, we have demonstrated a selective requirement of Mli1 for transformed stem cells but not in normal stem cells (Fig. 2d, e in the revised manuscript). We also observe this *in vivo* (Fig. R2a, sent separately to the reviewer).

Figure R2 sent to reviewers: Loss of Mli1 prevents the $\beta\text{-cat}^{\text{GOF}}$ -driven expansion of Lgr5-GFP⁺ intestinal stem cells.

a) Immunofluorescence for $\beta\text{-catenin}$ and Lgr5-GFP (green) on intestinal crypts of Mli1^{+/+}, Mli1^{-/-}, $\beta\text{-cat}^{\text{GOF}}$; Mli1^{-/-} and $\beta\text{-cat}^{\text{GOF}}$; Mli1^{+/+} mice at days 50 and day 20 after the induction of mutagenesis. Scalebars 25 μm , $\beta\text{-cat}^{\text{GOF}}$; Mli1^{+/+} 50 μm . **b)** LacZ tracing of mutant cells (blue), nuclear fast red counterstaining. Cropped from tracing images shown in Supplementary Fig. 2 and 3.

Lgr5-GFP⁺ mutant stem cells persist in the crypts of *Mll1*^{+/-}, *Mll1*^{-/-} and β -cat^{GOF}; *Mll1*^{-/-} mice, demonstrating that the loss of *Mll1* does not ablate stem cells at the crypt base (Fig. R2a, first three panels). The loss of *Mll1* prevents the expansion of the transformed β -cat^{GOF} stem cells, though: β -cat^{GOF}; *Mll1*^{-/-} stem cells persist at the crypt base but do not expand into tumors as it is observed for β -cat^{GOF}; *Mll1*^{+/-} intestines (Fig. R2a, third and fourth panel). In Supplementary Fig. 2 and 3, we have shown that LacZ⁺ *Mll1*-deficient stem cells give rise to LacZ⁺ mutant progeny (Fig. R2b, images cropped from the tracing images shown in Supplementary Fig. 2 and 3 of the manuscript). Of note, Francis Stewart and co-workers will soon submit a more detailed manuscript on their analyses of *Mll1* in normal intestinal stem cells (Goveas et al. 2020, *in preparation*). In the present manuscript, we demonstrate that *Mll1* is essential for the expansion of transformed β -cat^{GOF} stem cells into intestinal tumors. Our data suggest that *Mll1* is dispensable for the existence of functional intestinal stem cells in the crypt stem cell niches.

We now further address the role of the stem cell niche in maintaining *Mll1*-deficient β -cat^{GOF} stem cells. In addition to Wnt ligands produced by Paneth cells, the presence of the Bmp antagonist Noggin is essential for the maintenance and growth of intestinal stem cells *in vivo* and in organoid culture^{3, 11}. The RNA-seq of sorted β -cat^{GOF}; *Mll1*^{-/-} stem cells (Fig. 3b in the manuscript) reveals a decreased expression of the transcription factor *Gata4* (Fig. 3d, e in the manuscript) and a concomitant increase in the expression of the Bmp receptor ligand *Bmp4* in *Mll1*-deficient stem cells, which we also observe in β -cat^{GOF}; *Mll1*^{-/-} organoids compared to β -cat^{GOF}; *Mll1*^{+/-} controls (new Fig. 3f in the revised manuscript). The β -cat^{GOF}; *Mll1*^{-/-} organoids exhibit a decreased expression of stem cell genes (Fig. 2g in the manuscript) and increase the expression of the secretory differentiation markers *Mmp7* and *Muc2* (new Fig. 3f in the revised manuscript). We now write on page 10, lines 234-239:

Gata4/6 transcription factors had previously been reported to repress Bmp4, thereby maintaining colon cancer stemness¹². RT-PCR analysis of β -cat^{GOF}; *Mll1*^{-/-} organoids confirmed a decreased expression of Gata4 and revealed a concomitant upregulation of Bmp4 and the secretory differentiation markers Mmp7 and Muc2 (Fig. 3f), identifying *Mll1* as an upstream regulator of Gata4/Bmp4-controlled cancer stemness and differentiation.

To experimentally address our hypothesis that Bmp antagonists in the stem cell niche maintain stemness and *Lgr5* expression of β -cat^{GOF}; *Mll1*^{-/-} stem cells as long as the cells reside within the Noggin-rich niche (in organoid culture and *in vivo*), we modulated the niche in organoid culture. We applied the Bmp ligand *Bmp4* and withdrew the Bmp antagonist Noggin from β -cat^{GOF}; *Mll1*^{+/-} and β -cat^{GOF}; *Mll1*^{-/-} organoids and analysed their stem cell gene and differentiation marker expression, respectively. The activation of Bmp signalling by addition of *Bmp4* strongly reduced the expression of stem cell genes *Lgr5*, *Smoc2* and *Olfm4* in β -cat^{GOF}; *Mll1*^{+/-} organoids (Fig. R3a, sent to reviewer). The withdrawal of Noggin from β -cat^{GOF}; *Mll1*^{-/-} organoids, which endogenously produce *Bmp4* (Fig. 3f in the manuscript), further increased the expression of differentiation markers in these organoids (Fig. R3b, sent to reviewer). These data support our hypothesis on a role of the niche factor Noggin in preventing Bmp4-induced differentiation of β -cat^{GOF}; *Mll1*^{-/-} stem cells. The *Mll1*-dependent control of *Gata4/Bmp4* is dispensable for the maintenance of β -cat^{GOF}; *Mll1*^{-/-} stem cells in the Noggin-rich stem cell niche. Once leaving the Noggin-rich environment, however, the *Bmp4* produced by β -cat^{GOF}; *Mll1*^{-/-} stem cells is not antagonized by Noggin and increases epithelial Bmp signalling, which in turn represses *Lgr5*/stemness and promotes epithelial differentiation, as has been described by He et al. (2004)¹³ and Qi et al. (2017)¹⁴.

Figure R3 sent to reviewers: Bmp4 restrains stemness and promotes differentiation of $\beta\text{-cat}^{\text{GOF}}$ organoids.

a) mRNA expression of stem cell genes *Lgr5*, *Smoc2* and *Olfm4* in $\beta\text{-cat}^{\text{GOF}}; \text{Mll1}^{\text{+/-}}$ organoids cultured in ENR (control) and ER supplemented with 5ng/ml rBMP4 for 72h, n=4, unpaired t test. **b)** mRNA expression of secretory differentiation markers *Atoh1*, *Mmp7*, *Itf*, *Gob5* and *Muc2* in $\beta\text{-cat}^{\text{GOF}}; \text{Mll1}^{\text{-/-}}$ organoids cultured in ENR (control) and ER for 72h, n=3, unpaired t test.

In a tumor setting, the inhibition of Mll1 would likewise lead to a decrease in *Gata4/6* expression and consequent de-repression of *Bmp4*, inducing differentiation of Mll1-deficient colon cancer stem cells. The loss of Mll1 in our organoid and *in vivo* models prevents the acquisition of a tumorous phenotype. Hence, these models are unfit to analyse the benefit of Mll1 ablation in established tumors. We therefore analysed the role of MLL1 in human colon cancer cells cultured under non-adherent stem cell-enriching 3D sphere conditions and in xenografts. As we have shown in our manuscript, the knockdown of MLL1 induced the expression of secretory differentiation markers in human colon cancer spheres and xenografts (Supplementary Fig. 4n-o). MLL1 knockdown spheres and tumors exhibited an increased expression of the differentiation-associated p21 (Supplementary Fig. 4p, q and Fig. R4a sent to the reviewer). Inhibition of BMP signalling by Noggin treatment decreased phosphorylation of SMAD1/5/8 and reduced p21 levels in shMLL1 Ls174T sphere cells (Fig. R4a, quantifications on the right). BMP4 treatment of stem cell-enriched non-adherent sphere cultures of Ls174T cells induced differentiation of the human colon cancer cells (Fig. R4b, sent to the reviewer), as previously described¹⁵.

Figure R4 sent to reviewers: Bmp signalling causes loss of stemness and differentiation of Ls174T 3D sphere cells.

a) Western blot for p21, p SMAD1/5/8 and SMAD1/5/8 in untreated and 48h Noggin-treated control and 7d doxycycline-induced shMLL1 Ls174T sphere cells, tubulin as control for equal loading. Below: Quantifications of p21 and p-SMAD1/5/8 levels normalized to tubulin and SMAD1/5/8, respectively, in control, shMLL1 and Noggin-treated shMLL1 cells on the right, n=4, unpaired t test. **b)** Immunostaining for the differentiation marker CK20 (red) in control and 6d 100ng/ml BMP4-treated Ls174T secondary sphere cells. E-cadherin (yellow) stains cell borders, DAPI (blue) stains nuclei. Scale bars 25µm.

To sum up, we provide evidence for an essential role of Mll1 in β -cat^{GOF}-transformed intestinal stem cells, while Mll1 is dispensable for the viability and proliferation of Lgr5⁺ intestinal stem cells at homeostasis. Our data demonstrate that Mll1 sustains Wnt-induced colon cancer stemness by directly regulating stem cell gene expression and by controlling Gata4/6 transcription factors to antagonize epithelial Bmp4 production and stem cell differentiation.

Additional points have been mentioned by the editor: *“Similar concerns [on the selective requirement of Mll1 in cancer stem cells] were raised by Reviewer #2 when asking about Mll1-silencing effects on 2D versus 3D growing (point 3) and found the answer to this point was contradictory to the response to Reviewer #3 about the selective requirement of Mll1 for transformed stem cells only if not residing in the niche. She/he wondered whether attachment in 2D would be similar to the niche or what other differences between 2D/3D could explain the distinct dependencies on Mll1 in cancer versus normal stem cells.”*

Our answer to reviewer’s #2 point 3 is coherent with the answer we gave to reviewer #3 on the selective requirement of Mll1 for transformed stem cells only if not residing in the niche. We would like to point out that our analyses of stem cell expansion focus primarily on tumor initiation and that our organoid/*in vivo* systems and the 2D/3D sphere cultures of human colon cancer cells address two distinct aspects of intestinal tumors. The β -cat^{GOF}; Mll1^{flox} organoids and the *in vivo* tumor model provide information on the role of Mll1 in the initiation of oncogenic Wnt-driven intestinal cancer. The loss of Mll1 renders mutant β -cat^{GOF} stem cells incapable of expanding outside of the niche to form tumors, as outlined above (Fig. R2). Initially, progeny is produced from β -cat^{GOF}; Mll1^{-/-} stem cells, but at later stages is only rarely observed in crypts, which morphologically resemble wild-type crypts. Thus, a low proportion of β -cat^{GOF}; Mll1^{-/-} stem cells persists in the niche but these mutant stem cells do not transform into tumor-initiating cells. Of note, these persisting stem cells are still mutant for β -cat^{GOF}. Our analyses of Mll1^{flox} mice without oncogenic β -cat^{GOF} mutation, however, suggest that Mll1 is dispensable for the existence of functional intestinal stem cells at homeostasis (Fig. R2, Supplementary Fig. 3 in the revised manuscript). In established tumors the stem cells have already expanded independently of the intestinal niche architecture. As the loss of Mll1 prevents the β -cat^{GOF}-driven stem cell expansion and tumor formation, we cannot analyse the role of Mll1 in cancer stem cells in established tumors using the β -cat^{GOF}; Mll1^{-/-} model. To gain insights into the role of MLL1 in colon cancer stem cells in established tumors, we therefore analyse the effect of MLL1 depletion by inducible knockdown of MLL1 in 2D/3D sphere cultures and xenografts of human colon cancer cells. Non-adherent secondary sphere cultures in 3D are an established way to investigate cancer cell self-renewal and stemness^{1, 16}. In contrast, the adherent growth in 2D culture reveals the effect of MLL1 on the growth of (bulk) tumor cells of a more differentiated nature¹, but does not reflect normal stem cells or niche requirements. The effect of MLL1 depletion on the growth of 3D sphere cultures was more prominent than in 2D cultures, reflecting a role of MLL1 in the cancer stem cell population (Supplementary Fig. 4f, g in the revised manuscript).

In terms of a cancer therapy, our data suggest that targeting MLL1 is a promising way to induce differentiation of colon cancer stem cells, which represent a major problem in cancer therapy. Such an MLL1 inhibitor would need to be used as part of a combination therapy, since it does not efficiently deplete proliferation of the tumor bulk (represented by the 2D culture in Supplementary Fig. 4f, g).

Additional References to Reviewers

1. Zhu, Q. *et al.* The Wnt-Driven Mll1 Epigenome Regulates Salivary Gland and Head and Neck Cancer. *Cell Rep* **26**, 415-428 e415 (2019).
2. Merlos-Suarez, A. *et al.* The intestinal stem cell signature identifies colorectal cancer stem cells and predicts disease relapse. *Cell Stem Cell* **8**, 511-524 (2011).
3. Sato, T. *et al.* Single Lgr5 stem cells build crypt-villus structures in vitro without a mesenchymal niche. *Nature* **459**, 262-265 (2009).
4. Sato, T. *et al.* Paneth cells constitute the niche for Lgr5 stem cells in intestinal crypts. *Nature* **469**, 415-418 (2011).
5. Kim, T.H., Escudero, S. & Shivdasani, R.A. Intact function of Lgr5 receptor-expressing intestinal stem cells in the absence of Paneth cells. *Proc Natl Acad Sci U S A* **109**, 3932-3937 (2012).
6. Durand, A. *et al.* Functional intestinal stem cells after Paneth cell ablation induced by the loss of transcription factor Math1 (Atoh1). *Proc Natl Acad Sci U S A* **109**, 8965-8970 (2012).
7. van de Wetering, M. *et al.* The beta-catenin/TCF-4 complex imposes a crypt progenitor phenotype on colorectal cancer cells. *Cell* **111**, 241-250 (2002).
8. Schwitalla, S. *et al.* Intestinal tumorigenesis initiated by dedifferentiation and acquisition of stem-cell-like properties. *Cell* **152**, 25-38 (2013).
9. van Es, J.H. *et al.* Wnt signalling induces maturation of Paneth cells in intestinal crypts. *Nat Cell Biol* **7**, 381-386 (2005).
10. Sansom, O.J. *et al.* Loss of Apc in vivo immediately perturbs Wnt signaling, differentiation, and migration. *Genes Dev* **18**, 1385-1390 (2004).
11. Kosinski, C. *et al.* Gene expression patterns of human colon tops and basal crypts and BMP antagonists as intestinal stem cell niche factors. *Proc Natl Acad Sci U S A* **104**, 15418-15423 (2007).
12. Whissell, G. *et al.* The transcription factor GATA6 enables self-renewal of colon adenoma stem cells by repressing BMP gene expression. *Nat Cell Biol* **16**, 695-707 (2014).
13. He, X.C. *et al.* BMP signaling inhibits intestinal stem cell self-renewal through suppression of Wnt-beta-catenin signaling. *Nat Genet* **36**, 1117-1121 (2004).
14. Qi, Z. *et al.* BMP restricts stemness of intestinal Lgr5(+) stem cells by directly suppressing their signature genes. *Nat Commun* **8**, 13824 (2017).
15. Lombardo, Y. *et al.* Bone morphogenetic protein 4 induces differentiation of colorectal cancer stem cells and increases their response to chemotherapy in mice. *Gastroenterology* **140**, 297-309 (2011).
16. Vermeulen, L. *et al.* Single-cell cloning of colon cancer stem cells reveals a multi-lineage differentiation capacity. *Proc Natl Acad Sci U S A* **105**, 13427-13432 (2008).

REVIEWERS' COMMENTS

Reviewer #3 (Remarks to the Author):

Authors have now answered all my previous concerns.